# Impact of dual-layer solid-electrolyte interphase inhomogeneities on early-stage defect formation in Si electrodes

Chunguang Chen [1,2,9✉], Tao Zhou[3,4,9], Dmitri L. Danilov[1,2], Lu Gao[2], Svenja Benning[1,5], Nino Schön[1,5], Samuel Tardif [6], Hugh Simons [7], Florian Hausen [1,5], Tobias U. Schülli[3], R.-A. Eichel[1,5] & Peter H. L. Notten [1,2,8✉]

While intensive efforts have been devoted to studying the nature of the solid-electrolyte interphase (SEI), little attention has been paid to understanding its role in the mechanical failures of electrodes. Here we unveil the impact of SEI inhomogeneities on early-stage defect formation in Si electrodes. Buried under the SEI, these early-stage defects are inaccessible by most surface-probing techniques. With operando full field diffraction X-ray microscopy, we observe the formation of these defects in real time and connect their origin to a heterogeneous degree of lithiation. This heterogeneous lithiation is further correlated to inhomogeneities in topography and lithium-ion mobility in both the inner- and outer-SEI, thanks to a combination of operando atomic force microscopy, electrochemical strain microscopy and sputter-etched X-ray photoelectron spectroscopy. Our multi-modal study bridges observations across the multi-level interfaces (Si/Li$_x$Si/inner-SEI/outer-SEI), thus offering novel insights into the impact of SEI homogeneities on the structural stability of Si-based lithium-ion batteries.

[1] IEK-9, Forschungszentrum Jülich, D-52425 Jülich, Germany. [2] Eindhoven University of Technology, P.O. Box 5135600 MB Eindhoven, The Netherlands. [3] ID01 ESRF, CS 40220, F-38043 Grenoble Cedex 9, France. [4] Nanoscience and Technology Division, Argonne National Laboratory, Chicago 60439 IL, USA. [5] RWTH Aachen University, D-52074 Aachen, Germany. [6] Univ. Grenoble Alpes, CEA, IRIG-MEM, Grenoble 38000, France. [7] Department of Physics, Technical University of Denmark, 2800 Kongens Lyngby, Denmark. [8] University of Technology Sydney, Broadway, Sydney, NSW 2007, Australia. [9] These authors contributed equally: Chunguang Chen, Tao Zhou. ✉email: c.chen@fz-juelich.de; p.h.l.notten@tue.nl

L ithium-ion batteries (LIB) are nowadays the standard energy source for most of our modern portable electronics, as well as for newly emerging applications such as electrical vehicles and aircrafts[1]. Si is one of the most studied anode materials for LIB, due to its high energy density, mature industrial manufacturing, and abundance in the earth's crust[2–4]. In terms of theoretical specific capacity, Si can alloy with lithium up to $Li_{4.4}Si$ at high temperatures (415 °C), delivering a storage capacity of as high as 4200 mAh g$^{-1}$ [5,6]. $Li_{15}Si_4$ is formed at room temperature, with a capacity of 3579 mAh g$^{-1}$ [6–9]. The high alloying state is, however, accompanied by extreme volume changes of about 300%, which results in a rapid decay in storage capacity upon cycling due to mechanical fracture of the Si host material[6,8]. Hence, understanding the evolution of structural deformation during the (de)lithiation process is key to improving the cyclability of Si-based LIB.

Considerable efforts, both experimentally and in modeling, have been devoted to the characterization of the structural deformation in Si electrodes. Electron-microscope analysis indicated that the lithiation-induced expansion of crystalline Si preferentially occurs along the (110) crystallographic facets[10–12]. By using a substrate curvature technique, the stress and fracture energy evolution of thin-film Si anodes were measured as a function of (de)lithiation[13]. Upon lithiation, Si thin-film electrodes began to deform plastically, with a compressive stress of ~1200 MPa at a lithiation state of $Li_{0.4}Si$, followed by a decrease to ~450 MPa upon further lithiation to $Li_{3.75}Si$. During delithiation the compressive stress was reverted into tensile stress, peaking again at 1200 MPa for $Li_{0.33}Si$[13].

The stress evolution in Si nanoparticles was found to be different from that in thin-film Si electrodes by operando Raman Spectroscopy[14,15] and X-ray diffraction[15]. At the beginning of the initial lithiation process, the core of the nanoparticles was under tensile stress due to the presence of a native surface oxide layer, and to the outward expansion of the lithiated volume. Further lithiation inverted this into compressive stress as the lithiation front was trapped between the lithiated shell and the pristine Si core. Subsequent cycling of partially lithiated nanoparticles showed that while compressive stress was maintained on the core during delithiation, tensile and compressive stress were respectively applied during the re-lithiation of the amorphous shell and the lithiation of the pristine crystalline core material[15]. The difference in stress evolution between a curved reaction front in nanoparticles and a flat reaction front in thin-film electrodes was described elsewhere[16]. Besides, Si-based nanoparticle/thin-film anodes are shown to have a size- and thickness-dependent fracture behavior upon (de)lithiation[17,18].

The majority of these experiments were performed without any spatial resolution, yielding only averaged information on the evolution of structural deformation[10–18]. Local stress is therefore often underestimated and, in particular, the early stage of defect formation has mostly been neglected. Moreover, these experiments were performed in the presence of a liquid electrolyte, with the solid-electrolyte interphase (SEI) layer significantly hindering the investigations of mechanical failure mechanisms in the underlying Si electrodes. Therefore, spatially resolved studies that offer a better understanding of the correlation between the SEI and its impact on the structural stability of the underlying electrode are in urgent demand.

We study the evolution of structural deformation in Si electrodes, using a multimodal approach consisting of full-field diffraction X-ray microscopy (FFDXM), atomic force microscopy (AFM), electrochemical strain microscopy (ESM), and sputter-etched X-ray photoelectron spectroscopy (XPS). Thanks to the high sensitivity of operando FFDXM, we visualize defects at the pristine-Si/lithiated-Si interface (Fig. 1a) after lithiation to <0.2%

of the total capacity. We refer to them as early-stage defects due to their weak lattice deformation and low density. Quantitative analyses by three dimensional reciprocal space mapping (3D RSM) suggest that these defects are initially formed due to a heterogeneous degree of lithiation. Operando AFM, ex situ ESM and sputter-etched XPS measurements further trace the origin of the heterogeneous lithiation to inhomogeneities in the dual-layer SEI (Fig. 1b), in terms of topography and more importantly lithium-ion mobility. The significance of the early-stage defects is revealed after prolonged lithiation, as they are shown to be connected to the most deformed area or cracks in the Si electrode. Our multimodal study sheds light on the idea of improving the structural stability of Si-based LIB through minimization of inhomogeneities in the SEI. We further demonstrate this possibility by showing the complete absence of early-stage defects under similar cycling condition on samples coated with homogeneously deposited artificial SEI.

## Results

**Early-stage defect formation visualized by FFDXM.** FFDXM is a novel technique that combines the merits of hard X-ray diffraction (lattice deformation, penetration power) with microscopic imaging (spatial resolution, large field of view (FoV))[19,20]. The working principle of FFDXM is illustrated in Fig. 1a. As a diffraction method, FFDXM is sensitive to structural deformations with a lattice tilt resolution of 10$^{-2}$ mrad and strain resolution of 10$^{-4}$. Single-crystal Si wafers were used as working electrodes. The Si near the surface participates in the (de)lithiation process while the Si underneath serves as a sensor for structural deformations. At room temperature, the initial lithiation of Si was shown to be a two-phase phenomenon, with a sharp lithiation front separating the amorphized, lithiated-Si ($Li_xSi$) from the pristine, single-crystal Si. Upon delithiation Si remains amorphous, and any subsequent lithiation is a single-phase phenomenon characterized by a lithium concentration gradient across the material and a diffuse lithiation front[6,21]. The volume change accompanying the (de)lithiation process induces tensile (compressive) stress on the underlying Si (sensor), which can be monitored by shifts of the Bragg peak position in reciprocal space. Even for a small defect at its very early stage, the lattice deformation may extend laterally to a few micrometers, which can be precisely imaged with a 100 nm spatial resolution. The large FoV of about $100 \times 430$ μm$^2$ and an acquisition rate of 1 frame per second allows the surface area to be investigated under operando conditions, making FFDXM an ideal method to investigate structural deformation at the buried Si/$Li_xSi$ interface.

The Si electrode potential ($E$) is scanned by cyclic voltammetry (CV) between the open-circuit potential ($E_{ocp}$) and +5 mV at a scan rate of 5 mV s$^{-1}$. So-called time scans were recorded during the CV, which generates one dark-field snapshot every second at −0.03° off the Si (004) Bragg $\theta$ angle. At this offset angle, the integrated scattering intensity (Int) is sensitive to any deviation from the perfect single-crystal Si state and thus serves as a qualitative measure of structural deformations happening at the Si/$Li_xSi$ interface. In between each CV scan, a 3D RSM is performed, which allows quantitative information on the lattice strain-tilt to be measured simultaneously on the entire imaged area.

Figure 2 shows the result of operando FFDXM and the corresponding electrochemical data for the first three CV cycles. A small current peak was found near $E = 1.0$ V during the first CV cycle (Fig. 2a, encircled), which corresponds to the SEI formation by the decomposition of carbonaceous solvents and lithium salt[22]. The total Int curves of the entire region ($100 \times 430$ μm$^2$) in the first three cycles are shown in Supplementary Fig. 1.

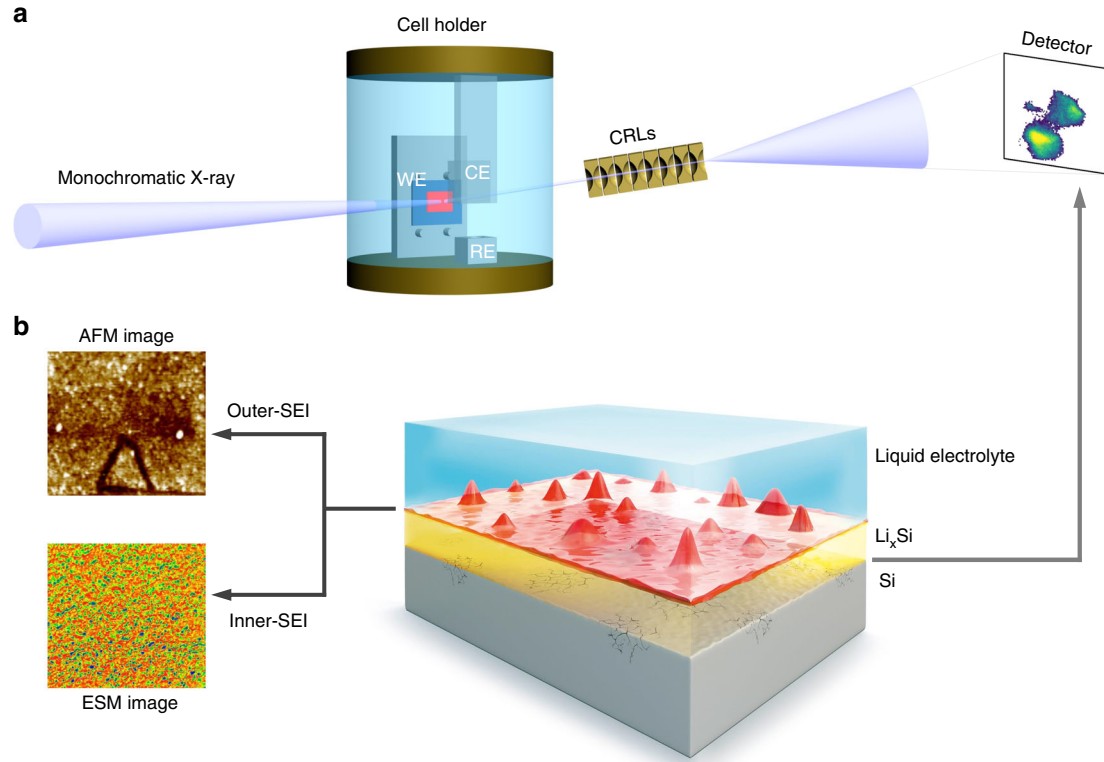

**Fig. 1 Schematic illustration of the multimodal study.** FFDXM working principle and operando electrochemical cell configuration (**a**). Single-crystal Si samples were prepared as the working electrodes (WE) and assembled in a custom-made cell with two lithium metal foils, acting as the reference (RE) and the counter electrode (CE). The cell is filled with 1 M LiPF$_6$ dissolved in ethylene carbonate/dimethyl carbonate (EC/DMC, 1:1 by volume). X-ray photons diffracted by the Si electrode are imaged by a set of compound refractive lenses (CRLs) onto an area detector. The exposure time per frame used to obtain the results in the present research is 1 s. The multilevel interfaces probed by FFDXM (Si/Li$_x$Si) as well as the combination of AFM, ESM, and XPS (inner-/outer-SEI) are schematically depicted in **b**.

No contrast was observed on the FFDXM images and Int remained 0 during the entire first cycle.

The SEI formation peak at 1.0 V in the 2nd cycle (Fig. 2b) was much less pronounced than in the first cycle. When $E$ reached its lowest value at +5 mV in the 2nd cycle, two defects appeared (Fig. 2d). These defects are structurally deformed Si and remained the only visible contrasts in the entire $100 \times 430\ \mu m^2$ FoV up through cycle 3. The 1:4.3 aspect ratio of the images (see Fig. 2d–n) is due to projection of the diffracted beam at shallow exit angle (13.4°). Selected FFDXM images are shown in Fig. 2e–n. For clarity, only the cropped ($5 \times 21.5\ \mu m^2$) area around the two defects is shown. The evolution of the two defects was analyzed by integrating separately the scattered Int in the two regions, denoted as Area I and II in Fig. 2i. It is evident that these two defects behave differently under the same cycling condition (Int curves in Fig. 2b, c). The current turned positive at point g (Fig. 2b) in the 2nd cycle, indicating the start of the delithiation process (red background). For Area I, most of the initial deformation was mitigated by spontaneous stress relaxation even before delithiation, which explains the weak contrast in Fig. 2g. Int of Area I at point i was only slightly lower compared with that at point h (purple curve of Fig. 2b), suggesting a significant amount of residual stress remained at the end of the 2nd cycle. In contrast, Area II was slow in relaxing the initial stress (strong contrast in Fig. 2g) but was able to spontaneously relax most of the residual stress after the stress reversal (much lower Int at point i compared with point h, orange curve in Fig. 2b).

A small depression in Int was observed at the start of the 3rd lithiation (point k in Supplementary Fig. 1c). This can be understood as the residual stress from the previous cycle being mitigated by another stress reversal[13]. The depression was more significant for Area I (point k in Fig. 2c) because most of the residual stress had been spontaneously relaxed for Area II at the end of the previous cycle. The much weaker contrast in the corresponding FFDXM image (Fig. 2k) indicates that the early-stage defects are mainly elastic, as most of the deformed Si was, at this point, restored to normal. The heterogeneous evolution of the two defects is very well visible and serves as a good example why spatially resolved techniques are required to fully understand the failure mechanism in such systems where inhomogeneous behavior is to be expected. The complete evolution of the two defects during cycle 2 and 3 can be found in Supplementary Movie 1 and 2, respectively.

3D RSM was performed at the end of cycle 3 to quantitatively characterize the two defects. Figure 2n shows the raw FFDXM image at the end of the 3rd cycle, which corresponds to deformed Si diffracting specifically at −0.03° off the Bragg peak. Figure 3a is the processed 3D RSM of the same area. The defects appear to be larger because the entire deformed area is shown. The main contrast is dominated by the lattice tilt, which is found to be as large as 0.03°. The lattices were tilted inwards, indicating a smaller lattice parameter at the center of the defects, and hence a lower degree of lithiation in the defects than in the surrounding area after delithiation (Fig. 3b). In other words, these early-stage defects were formed as a result of heterogeneous lithiation. The elongated shape of the two defects is again due to the projection factor at shallow exit angle (13.4°). The length scale of the defects was otherwise close to 4 μm in all directions in the surface plane.

**Fig. 2 Evolution of the early-stage defects from cycle 1 to 3.** Potential (E) and current (I) of the 1st cycle (**a**). The E, I, and the integrated scattered intensity (Int) of the 2nd (**b**) and 3rd (**c**) cycle are shown separately for Area I and II. The light green and light red background in **a**–**c** illustrate the current for lithiation (negative currents) and delithiation (positive currents), respectively. **d** shows the complete FoV of $100 \times 430\ \mu m^2$. **e**–**n** show the evolution of the observed defects at a higher magnification. The labels of **e**–**n** match the marked annotations in **b** and **c**, corresponding to the instants of time at which the FFDXM images were taken.

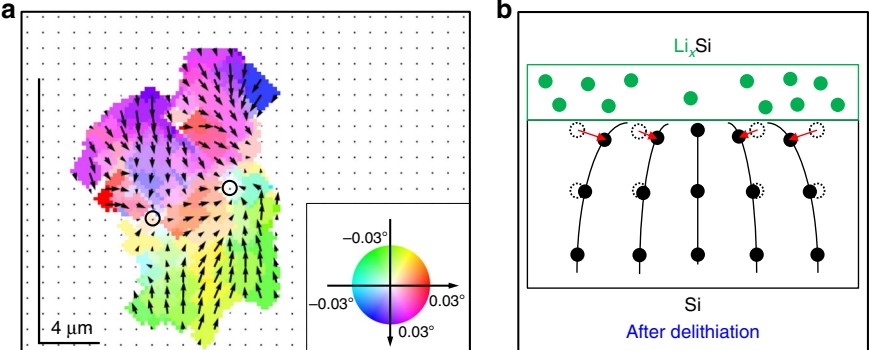

**Fig. 3 Quantitative analysis of the early-stage defects.** Result of 3D RSM (**a**). The direction and magnitude of the lattice tilt are expressed both by false colors (the reference color wheel is found in the inset) and arrows. The lattice tilt is the weakest at the center of the two defects, as indicated by the two circles in **a**. Cross-sectional schematic representation of the tilted lattice (**b**). The lattice is tilted toward the center of the defect, which is attributed to a lower lithium concentration in the defective area after delithiation.

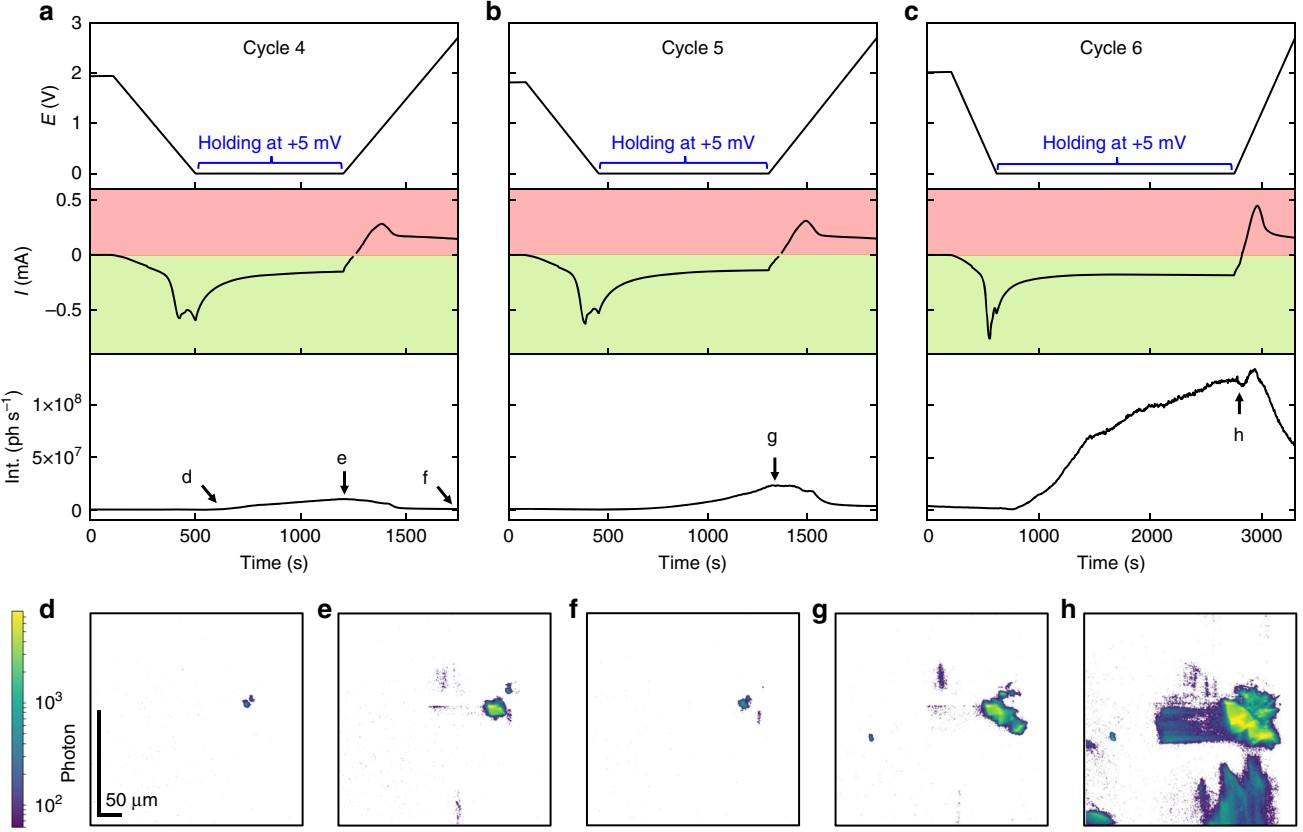

**Fig. 4 Defect evolution from cycle 4 to 6.** *E, I,* and Int curves of cycle 4 (**a**), 5 (**b**), and 6 (**c**). The light green and light red background illustrate the current range for lithiation (negative currents) and delithiation (positive currents), respectively. The labels of **d–h** match the marked annotations in **a–c**, corresponding to the instants of time at which the FFDXM images were taken.

It should be noted that due to the short period of time that the electrode potential was below the lithiation potential, the amount of incorporated lithium during each cycle was very limited, close to 0.1–0.2% of the full capacity. The lattice strain normal to the surface in the defective area is below the detection limit of $10^{-4}$, with stress being relaxed mainly through lattice tilts. This observation thus sheds light on the very early stage of defect formation where the average stress value on the entire sample is negligible and the defect density is low (<100 ppm). For discussion on the sensitivity of the FFDXM, the readers are kindly referred to Supplementary Fig. 2 and Supplementary Note 1.

Figure 4 shows the *E, I,* and Int curves for cycle 4, 5, and 6 as well as the corresponding FFDXM images. The entire $100 \times 430$ $\mu m^2$ FoV is shown. The two individual defects (Fig. 4d) merged into a large single defect at the end of the 4th lithiation cycle (Fig. 4e). New defects were also formed, as a result of prolonged lithiation by holding the electrode potential at +5 mV (Fig. 4e). These newly formed defects were, at this point, mainly elastic as confirmed by the vanished contrast after delithiation (Fig. 4f). At the end of the 6th cycle, deformed Si was observed to cover consistently ~20% of the sample surface area (Supplementary Fig. 3). We note that the two early-stage defects, despite being barely noticeable when initially formed (Fig. 2d), were responsible for the strongest structural defect (Fig. 4h) within the FoV, both in terms of physical size and amplitude of deformation. For discussion on the evolution of the defect density upon cycling, the readers are kindly referred to Supplementary Fig. 3.

**Operando AFM studies of the SEI layer.** Operando FFDXM offers an unobstructed view to the defect formation in the buried Si/Li$_x$Si interface, which is inaccessible to most surface probing techniques. However, as a diffraction method it is insensitive to the amorphous SEI layer above. Operando study of the SEI layer is equally important as it governs the lithium-ion transport in and out of the Si electrodes. Furthermore, the early-stage defects (Fig. 2d) were only observed after the initial formation of the SEI layer, which points to a possible correlation. To complement the FFDXM measurements, operando AFM was performed on the same batch of Si electrodes. The high surface sensitivity of the AFM enables direct observation of the SEI formation at the surface of pristine single-crystal Si electrodes. The experimental setup for operando AFM is schematically shown in Supplementary Fig. 4a. Peak-force quantitative-nano-mechanical (PF-QNM) mode[23] with a low contact force of 80 nN was used to minimize the disturbance of the AFM tip on the SEI formation.

Topography of the pristine single-crystal Si electrode at $E_{ocp}$ (Supplementary Fig. 5) reveals a typical smooth Si surface with a root mean square (RMS) roughness of 1.2 nm. Operando topographical measurements (Fig. 5a) of the same Si electrode were performed by scanning the electrode potential from $E_{ocp}$ (~3.0 V) to +5 mV with a scan rate of 7 mV s$^{-1}$ while moving simultaneously the AFM tip in the direction indicated by the black arrow. The operation principles for operando AFM is shown in Supplementary Fig. 4b–e. The corresponding current–voltage curve is shown in Fig. 5a. A SEI-free Si surface was observed up until $E = 1$ V. The surface became rougher starting at 0.75 V, signaling the beginning of the SEI formation. This was also evident from the surface height profile (Fig. 5q) generated by subtracting the height of the pristine Si surface (green line in Supplementary Fig. 5) from that of the operando measurements (green line in Fig. 5a). It can be seen that SEI

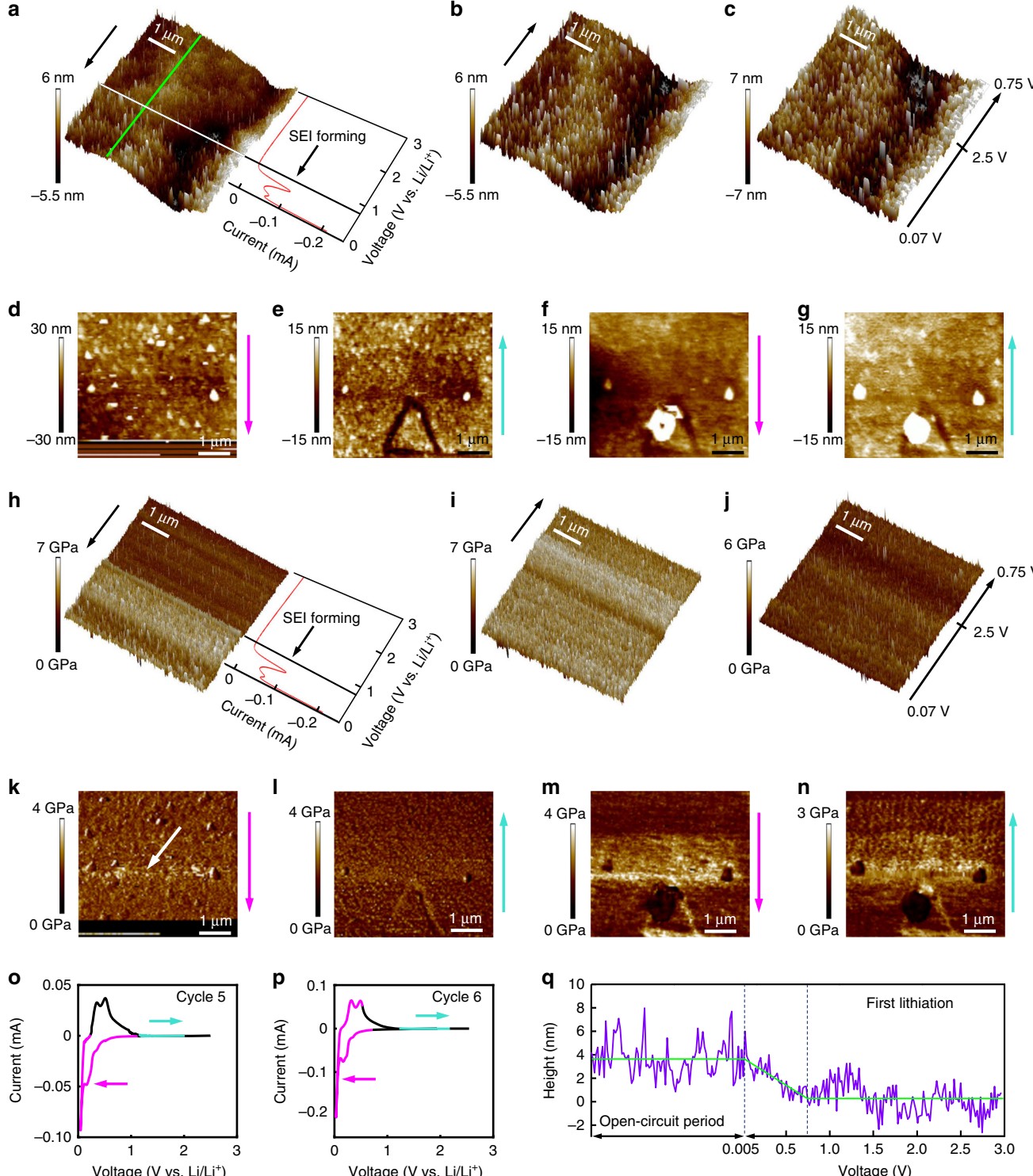

**Fig. 5 Operando AFM measurements on the single-crystal Si electrode.** Topographical map and current–voltage curve during the first CV scan from $E_{ocp}$ to the cut-off potential of +5 mV at a scan rate of 7 mV s$^{-1}$ (**a**), during the resting period (**b**), during the CV scan back to 2.5 V and down to 0.75 V (**c**). **d** and **e** are the topographic maps during the 5th CV cycle. **f** and **g** are the topographic maps during the 6th CV cycle. The current and voltage for **d**, **e** and for **f**, **g** are indicated respectively by the colored curve in **o** and **p**. The colored arrows to the right of each map indicate the direction of the topographic scans (**d–g**). Also shown are results of Young's modulus measurements. The Young's modulus maps in **h–n** are taken at the same time as the topographic maps in **a–g**, respectively. The missing data in **d** and **k** are caused by surface fluctuations during the crack formation. **q** shows the height profile of the surface during the first lithiation and subsequent open-circuit period.

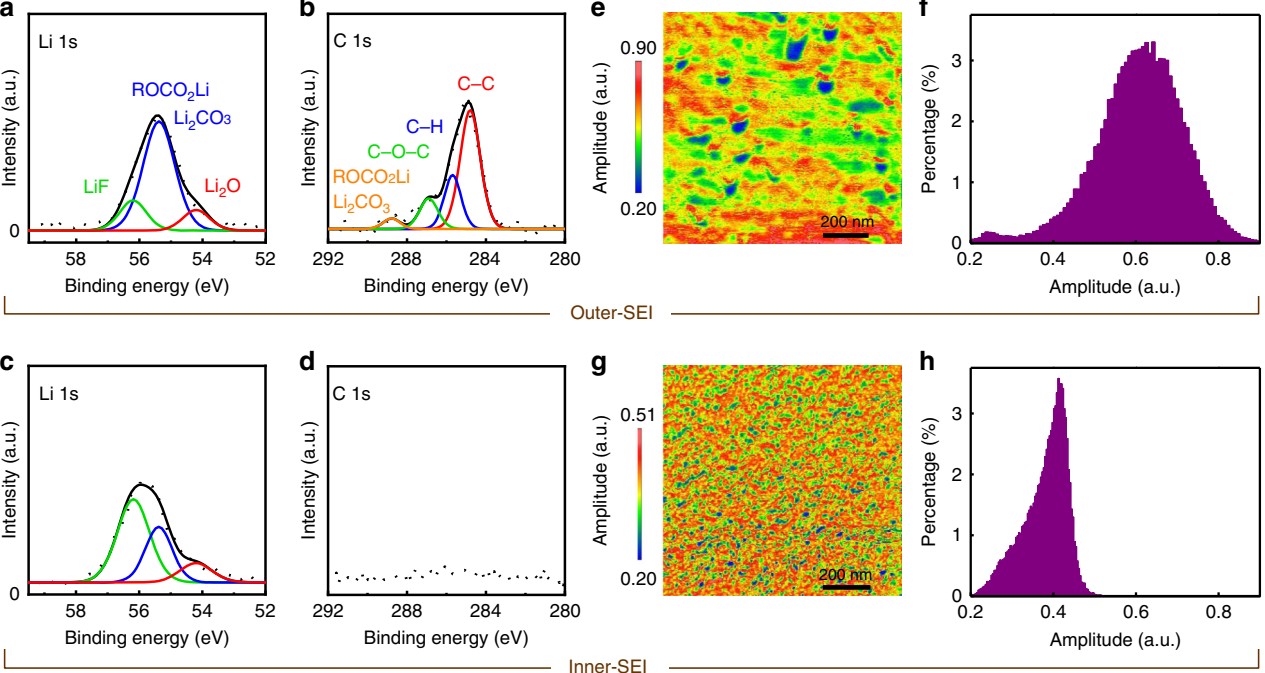

**Fig. 6 Composition and lithium-ion mobility of the dual-layer SEI.** Li 1s and C 1s sputter-etched XPS spectra of the Si electrode after the 6th cycle. The spectra in **a** and **b** were obtained without sputter-etching the SEI surface and represent the chemical information of the outer-SEI layer. The spectra in **c** and **d** were recorded after about 10 s of sputter-etching and represent the chemical information of the inner-SEI layer. ESM amplitude map (**e**) and corresponding histogram (**f**) of the Si electrode after the 6th cycle. ESM amplitude map (**g**) and corresponding histogram (**h**) of another Si electrode, after being scanned to +5 mV during its first CV cycle at a scan rate of 7 mV s$^{-1}$. The center contact resonance frequency of the tip-sample system was 290 kHz for the two ESM measurements.

started to develop when the sample was scanned at the SEI forming potential of 0.75 V. The current was switched off as soon as the electrode potential reached +5 mV. During the subsequent open-circuit period a complete topographical image was taken (Fig. 5b). The result shows that the surface was covered by an inhomogeneous SEI layer with an RMS roughness of 2.4 nm. Figure 5c shows the topographical AFM image during the scan back to 2.5 V and then down to 0.75 V.

To mimic the same conditions as in the FFDXM experiments, more CV cycles were applied. The SEI formation peak was not so pronounced in subsequent cycles as compared with the first one, consistent with the FFDXM observation. A triangular-shaped crack (Fig. 5e) was observed while scanning the electrode potential from +5 mV back to 2.5 V during the 5th cycle. The scanning periods for Fig. 5d, e are indicated by the magenta and turquoise curve in Fig. 5o, respectively. Due to this crack formation, the contact of the AFM tip with sample surface was lost for a brief period of time, as recognized by the blurry region in the lower part of Fig. 5d. The sample was further scanned for the 6th cycle. The scanning periods for Fig. 5f, g are indicated by the magenta and turquoise curve in Fig. 5p, respectively. New SEI was formed, as indicated by the bright contrast in Fig. 5f, g. In particular, a large bright spot was observed on top of the triangle-shaped cracked area found in Fig. 5e.

To understand the origin of the crack formation, Young's modulus or the stiffness of the electrode surface was measured while performing operando topographical measurements in the PF-QNM mode. Figure 5h–j shows the operando Young's modulus map of the Si electrode surface during the first CV cycle, measured simultaneously as the topographical images shown in Fig. 5a–c, respectively. Figure 5h clearly shows that the sample surface became harder when the potential reached the SEI forming potential at 0.75 V. Figure 5i shows that the entire

sample surface was covered by a layer of hard SEI materials during the open-circuit period. When the Si electrode was scanned back from 5 mV to 2.5 V, the sample surface subsequently became softer (Fig. 5j), the reason of which will be discussed in the next section.

The stiffness of the sample surface continued to decrease in the subsequent cycles. Figure 5k–n shows the Young's modulus maps of the 5th and 6th cycle, measured simultaneously as the topographical images shown in Fig. 5d–g, respectively. The crack was also clearly visible in the Young's modulus maps. Local hardening was observed at the upper corner of the triangular-shaped crack (Fig. 5m). The hardened areas manifested themselves as brighter contrast on the Young's modulus maps and were observed even before the cracking in Fig. 5k (white arrow). The connection between the AFM observations (local hardening to crack) and the FFDXM observations (early-stage to large-scale defects) will be discussed later. As the crack broke the SEI layer, new SEI was formed on freshly exposed Si in contact with the liquid electrolyte. It is worth mentioning that the newly formed SEI contains a high ratio of soft materials as indicated by the darker contrast on the Young's modulus maps (Fig. 5m, n).

**Inhomogeneity of lithium-ion mobility in dual-layer SEI.** From the operando Young's modulus maps, it can be inferred that the SEI layer formed during the 1st lithiation (Fig. 5i) was harder than that formed during subsequent cycles (e.g., Fig. 5j). This is because of the duality nature of the SEI. Similar to what was found on carbon-based electrodes[24–27], the SEI on Si also has a dual-layer structure, composed of an inner- and outer-SEI layer[28–30]. The so-called inner SEI mainly consists of inorganic species which are dense, thin, and relatively stable[24,31,32]. A so-called outer-SEI layer is formed on top of the inner-SEI layer, and

is mainly composed of organic material, which continues to grow upon cycling[24,28].

Sputter-etched XPS analyses have been performed to study the chemistry of the inner- and outer-SEI layers. Figure 6a–d shows the Li 1s and C 1s spectra of the dismantled sample (rinsed by DMC and dried) after the 6th CV cycle. The spectra in Fig. 6a, b are collected without sputter-etching the surface and are therefore representative of the outer-SEI layer. The result (Li 1s spectra, Fig. 6a) confirmed that the outer SEI mainly comprised of organic species, such as $ROCO_2Li$ (where R is a low-molecular-weight alkyl group)[33–35]. Figure 6b shows intensive peaks of multiple carbon-based species in the C 1s spectra. The predominant presence of organic materials leads to a softer outer-SEI layer as observed in the Young's modulus maps (Fig. 5j–n). Small amounts of LiF and $Li_2O$ were also observed in the outer-SEI layer, albeit significantly less than $ROCO_2Li$ (Fig. 6a). Figure 6c, d shows the spectra collected after about 10 s of sputter-etching and are representative of the inner-SEI layer. In sharp contrast, inorganic LiF was found to be the dominant component (Fig. 6c) with a significantly weaker presence of the carbon species (Fig. 6d). The result explains the origin of the harder inner-SEI layer as observed in the Young's modulus map (Fig. 5i).

Result of 3D RSM (Fig. 3) indicated heterogeneous lithiation to be responsible for the formation of the early-stage defects. The heterogeneous lithiation is likely to be caused by an inhomogeneous conduction of the lithium-ion in the SEI layer. To verify this, ex situ ESM was employed on the Si electrode after it was dismantled at the end of the 6th CV cycle. The electrode surface was subsequently rinsed by DMC and dried. The ESM amplitude is a measurement of the electrical-field-induced surface strain and is proportional to the lithium-ion mobility[36–38]. Figure 6e, f shows the ESM amplitude map and corresponding histogram. The ESM map in Fig. 6e shows a strong variation in lithium-ion mobility with some areas of the sample conducting lithium-ion more than four times less than some other areas. It is worth noting that the inhomogeneous lithium-ion mobility observed in Fig. 6e is characteristic of the outer-SEI layer. Ex situ ESM was also performed on the inner-SEI layer. This latter was made possible by dismantling another Si electrode after it was scanned to +5 mV in its 1st cycle. The Young's modulus measurement in Fig. 5i indicated that the Si electrode at this stage was only covered by the inner SEI. Figure 6g, h shows the ESM amplitude map and corresponding histogram. The results revealed that the lithium-ion mobility of the inner SEI was also inhomogeneous, albeit with a weaker variation.

## Discussion
The impact of the SEI on the structural deformation in Si electrodes was studied by a combination of operando and ex situ techniques. For operando studies, CV with a relatively fast scan rate of 5 mV s$^{-1}$ was used to intentionally limit the amount of lithiation during each cycle. This allowed us to observe the formation of defects at a mere 0.1–0.2% of the total lithiation capacity of Si. We refer to them as early-stage defects because of their low density (<100 ppm, Supplementary Fig. 3) and weak amplitude of deformation (Fig. 3a), making them essentially invisible to nonlocal techniques that average over the entire sample (Supplementary Fig. 2). Spontaneous relaxation was observed both during lithiation and delithiation, and the deformation around the early-stage defects was found to be mainly elastic (Fig. 2k). The significance of these early-stage defects was later revealed by prolonging the potential holding at +5 mV, during which both the defect density and their amplitude of deformation gradually increased. It was shown that, of the ~20% surface area (Supplementary Fig. 3f) covered by large-

scale defects, the most deformed part was evolved from the early-stage defects. In other words, suppressing the formation of early-stage defects is essential to improving the structural stability of Si electrodes upon cycling.

3D RSM result at the end of cycle 3 shows the lattice in the defective area to be tilting inwards (Fig. 3a), indicating locally a lower degree of lithiation compared with the surrounding area. This heterogeneous lithiation is likely to be caused by two major factors: inhomogeneities in the thickness of the lithium-ion-conducting SEI layer (longer transportation time for thicker regions) and in the mobility of the lithium-ion (less conducting for lower mobility). Both were supported by the present study. On one hand, operando topographic measurement showed a twofold increase in RMS roughness (standard deviation of layer thickness) in the initially formed SEI (Fig. 5b). On the other hand, ex situ ESM (Fig. 6e) result revealed a strong variation in lithium-ion mobility of the SEI layer, with some areas of the sample conducting lithium-ion more than four times less than other areas. These inhomogeneities inevitably led to a heterogeneous degree of lithiation, which in turn resulted in the lattice deformation as seen by FFDXM. To further confirm this correlation, we have performed FFDXM experiments on the same batch of Si samples coated with a homogeneous artificial SEI ($Li_4Ti_5O_{12}$-$Li_3PO_4$). Indeed, no defects were ever observed under similar cycling conditions. This last result can be found in Supplementary Fig. 6.

The SEI formed on Si is known to have a dual-layer structure, composed of a softer (Fig. 5j–n) and mainly organic (Fig. 6a, b) outer-SEI layer on top of a harder (Fig. 5i) and mainly inorganic (Fig. 6c, d) inner-SEI layer. We note that the average lithium-ion mobility of the outer-SEI layer (Fig. 6f) is ~1.5 times that of the inner one (Fig. 6h), which is explained by a higher lithium-ion conductivity for $Li_2CO_3$ and $ROCO_2Li$ (main components of the outer SEI) than for LiF (main component of the inner SEI)[39–42]. This indicates that the lithium-ion transport through the inner SEI is the major limiting factor in the case of Si electrodes. Optimizing the thickness and roughness of the inner SEI is thus more rewarding than tuning those of the outer one. The properties of the inner SEI are known to be affected by the surface of the electrode, due to their direct contact. As is the case for this study, the presence of native silicon oxide promotes the formation of a less rough and thinner SEI[43,44]. The use of rigid Si wafers with atomically flat surface also helps reduce the roughness of the SEI layer, which is favorable for observing the defects in their early stage.

Although operando FFDXM and AFM were not performed simultaneously on the same sample, the two observations were closely connected. The early-stage defects first appeared during the 2nd cycle in the FFDXM experiments (Fig. 2b, d). In the meantime, local (i.e., inhomogeneous) hardening of the Si electrode was also observed during the 2nd cycle in the AFM experiment. The local hardening manifests itself as brighter contrast in the Young's modulus map (black square, Supplementary Fig. 7c) and is explained by a local variation in the chemical composition. More importantly, FFDXM showed that after extensive cycling, the part where the Si lattice was most deformed lies on top of the early-stage defects (Fig. 4). Similarly, the cracking of the Si electrode observed by AFM (Fig. 5k–n) is also tied to the local hardened area. It is worth noting that the mismatch in spatial resolution and FoV between the FFDXM and AFM is in fact welcomed. The deformation in the lattice often extends much further than the physical size of the defects. The 100-nm spatial resolution of FFDXM is adequate for the investigation of the nature of the defects because their lattice deformation has the size of roughly $4 \times 4 \, \mu m^2$ (Fig. 3a). Its large FoV allowed us to detect those sparsely distributed defects when they

were initially formed. AFM, on the other hand, measures the physical size of the defects. Its finer spatial resolution (20 nm) is required to resolve the crack ($1 \times 1$ μm$^2$) and the topographic inhomogeneities ($0.1 \times 0.1$ μm$^2$, Fig. 5).

Our multimodal study bridges the gap between conventional X-ray diffraction, sensitive to structural deformation but not in its early stage (i.e., <100 ppm) and surface probing microscopy which measures topography, stiffness, lithium-ion mobility but not in buried layers. It reveals the presence of the early-stage defects and their ominous connection to the large-scale deformation causing mechanical failures in the electrodes. We note that the onset of the early-stage defects and their evolution have been consistently observed over repeated experiments, and under various cycling conditions (Supplementary Fig. 8). The present study allows us to construct a model (Supplementary Movie 3) that highlights the inhomogeneities in the dual-layer SEI and their correlation to the structural deformation inside Si electrodes in liquid electrolyte. This correlation may, for instance, explain the unexpected yet consistent crack observations in extremely thin (20 nm) Si film electrodes[18,45], despite a theoretical critical thickness of 100 nm[18,46]. More generally, we hope to accentuate the importance of minimizing the inhomogeneities in the SEI (natural or artificial) on improving the structural stability of Si-based LIB.

## Methods

**Electrodes preparation**. Single-crystal Si electrodes were fabricated by thermal evaporating 200 nm Cu on one side of highly doped, double-side polished, (100) oriented Si wafers (200 μm thick, UniversityWafer), acting as the current collector. The rigidness of the thick Si substrates is necessary for high-precision measurements of the structural deformation. Free standing thin films are not recommended as they are often bent (inherent lattice tilt) due to the absence of a rigid support, and easily strained (inherent lattice strain) by adhesives. The Si near the surface participates in the (de)lithiation process and is subsequently amorphized, the dark-field contrast in FFDXM thus reflects structural changes in single-crystal pristine Si just below the Si/Li$_x$Si interface.

**Operando FFDXM**. FFDXM experiments were carried out on the ID01 beamline at the European synchrotron radiation facilities (ESRF) in Grenoble, France. The size of the Si samples was $7 \times 5$ mm$^2$. The custom-made 3-electrode electrochemistry cell is controlled by a potentiostat (Biologic, SP300). 1 M LiPF$_6$ dissolved in EC/DMC (1:1 in volume ratio) was used as electrolyte. All potentials are given with respect to the Li/Li$^+$ reference electrode.

FFDXM is a novel synchrotron technique that combines X-ray diffraction with spatially resolved microscopy. The basic principles of FFDXM time-scan and 3D RSM are schematically shown in Supplementary Fig. 9. The Si samples were mounted vertically so that the surface normal lies in the horizontal scattering plane (the $X$–$Y$ plane, Supplementary Fig. 9a). The illuminated sample area is 980 ($H$) × 240 ($V$) μm. The pre-focusing by the in vacuum Beryllium transfocator at 62-m upstream of the sample allows a high incoming photon flux to be achieved with a near parallel beam (convergence < 0.03 mrad). A set of CRLs (SU-8, made by the Karlsruhe Institute of Technology, 6 μm apex radius of curvature[47]) was mounted behind the sample to image the diffracted beam. For each point of data acquisition, a real space image (raw FFDXM image) was acquired on the area detector (Andor Zyla 5.5 sCMOS, 6.5 μm pixel size, 2560 × 2160 pixels), which is direct fiber-coupled to a 15 μm of Gadox scintillator. The X-ray energy for the experiments was chosen to be 19.7 keV to reduce absorption from the operando electrochemical cell and the electrolyte. For the operando measurements, the so-called time scans were performed at a sample angle $-0.03°$ off the $\theta$ angle for the (004) reflection (tilted by the phi motor). Supplementary Fig. 9b shows the integrated Int of a rocking curve around the (004) Si reflection. The full width at half maximum is broadened by the bandwidth of the multilayer monochromator ($\Delta E/E = 3 \times 10^{-3}$). At $-0.03°$ off the Bragg angle (marked by the arrow in Supplementary Fig. 9b), the technique is highly sensitive to any deformation of Si, deviating from the perfect single-crystal structure. For the 3D RSM, the sample was scanned in three orthogonal directions in the reciprocal space, given respectively by the phi rocking, the psi rocking, and the **Q** (the momentum transfer) direction (Supplementary Fig. 9c). The two sample rocking motions allow the lattice tilt to be mapped at a resolution of $10^{-2}$ mrad, whereas the radial motion in the **Q** direction (achieved via a $\theta$–$2\theta$ motion by moving simultaneously the sample phi, the detector, and the lenses) allows the lattice strain to be mapped at a resolution of $10^{-4}$.

**Operando AFM**. An AFM (Bruker, Santa Barbara, USA, Dimension Icon Microscope) setup, operating inside an Argon-filled glove box (H$_2$O < 0.1 ppm, O$_2$ < 0.1

ppm, MBraun, Stratham, USA), was used to perform the operando AFM measurements. The design of the in-house made cell holder for the AFM measurements is shown in Supplementary Fig. 4a. The Si single-crystal working electrode is assembled in the AFM cell with two lithium metal foils, acting as the RE and the CE and 1 M LiPF$_6$ dissolved in 1:1 EC/DMC as the electrolyte. PF-QNM mode[23] was employed to carry out the topographical investigations. Tapping mode cantilevers (NCHR, NanoSensors) made of doped silicon with a nominal spring constant of 64 nN nm$^{-1}$ were used as force transducer. All images have been captured with a tip-force of 80 nN at a scan rate of 0.5 Hz in the trace–retrace mode. The direction of the tip movement is indicated in each image. All AFM images are recorded with a pixel density of 256 data points per line.

**Electrochemical strain microscopy (ESM)**. For lithium-ion movability detections, the AFM was operated in the ESM mode, which is effective to qualitatively visualize local variations[36,37]. ESM, conducted in the contact mode, is a novel AFM technique. It concentrates a periodic electrical field with the same frequency as the contact resonance frequency at the conductive AFM tip within a radius of about 10 nm[36,37]. The induced electrical field then moves lithium ions toward or away from the electrode surface[38]. This results in local electrochemical surface strain, which is proportional to the local lithium-ion mobility[36,37]. In the ESM mapping mode, the cantilever was changed to a conductive one with a platinum/iridium coating (PPP-EFM, Nanosensors) and a nominal spring constant of 2.8 N m$^{-1}$. The contact resonance frequency and amplitude were tracked with a phase-locked loop (HF2LI, Zurich Instruments, Switzerland). This ensures that frequency shifts during the measurement are tracked and measurements are always on resonance. The center frequency is given in the various ESM-related figure captions. To ensure a stable tip-sample interaction, a slow scanning frequency of about 1.5 Hz was applied with 256 data points per line and a drive amplitude of 2 V. The two ESM measurements were performed in trace and retrace mode with the same parameters by using identical tip.

**Sputter-etched XPS**. XPS spectra were obtained with a ThermoScientific K-Alpha instrument equipped with a monochromatic X-ray source (Al Kα = 1486.6 eV). The Si sample after 6 CV cycles was etched at an input power of 300 W, a gas pressure of 30 mTorr, and a gas flow of 8% oxygen content. All samples were transferred in a shuttle transfer system under Ar from the glove box (Unilab PRO, MBraun) to the FFDXM, AFM, ESM, and XPS equipment.

## Data availability

The data that support the graphs within this paper and other findings of this study are available from the corresponding author upon reasonable request.

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

## Acknowledgements

The authors are grateful to the ESRF for providing beam time at ID01. C.C. acknowledges the Chinese Scholarship Council (Grant No. 201506020080) for partially financial support of this work. T.Z. acknowledges support from the Center for Nanoscale Materials, a U.S. Department of Energy Office of Science User Facility under Contract No. DE-AC02-06CH11357. T.Z. would like to thank Jakub Drnec and Manuel Marechal for the helpful discussions.

## Author contributions

C.C. and T.Z. contributed equally to this work. C.C. and T.Z. conceived the idea and conducted most of the experimental work. C.C. has fabricated the samples. C.C. has analyzed the samples by operando AFM, ESM, and XPS with the assistance of S.B., N.S., and L.G. C.C., T.Z., D.L.D., and P.H.L.N. composed the manuscript with the comments from all authors. T.Z. analyzed the FFDXM data. D.L.D., S.T., H.S., F.H., T.U.S., R.-A.E., and P.H.L.N. contributed to the scientific discussions.

## Competing interests

The authors declare no competing interests.
