## [Peer Review File · Nature Communications]

Reviewers' Comments:

Reviewer #1:

Remarks to the Author:

Si is believed to be the next generation anode material for Li-ion batteries and fundamental understanding of the fading mechanism is essential for developing better Si anodes. With recent advances in Si structure design and performance improvement, understanding and improving SEI resiliency are currently of growing interest. From this point, the work potentially can have a broad impact. The authors in this paper used operando FFDXM, AFM and ESM to help understand the fading mechanism of Si structure and interphase evolution with cycling. It is unique capability, yet I don't find enough new knowledge beyond expectation to make it publishable in Nature Communications. Below please find my detail comments.

First, I found it quite hard to understand some of the authors claims. What do the authors mean by early stage detection? The FFDXM signals start to show up in the 2nd cycle and after full lithiation. While Si has experienced expansion in the first lithiation and shrinkage in the first delithiation. What do the authors mean by "defects", "structural defects", what defects and defect of what?

Second, the definition of SEI to me is the solid electrolyte interphase layer on top of the anode that has the specific function. With that in mind, I tend to believe the SEI is SEI after the 1st full lithiation process. It is good to study the formation process of the SEI. Yet, I think it is more appropriate to call "pre-SEI" or "early SEI" for what detected during the first lithiation process in the operando AFM and ESM characterization.

Third, it is general belief that the SEI mostly forms in the first cycle and some in the 2nd cycle. The authors also showed support of SEI formation in 1st cycle in AFM measurement. Si structure change also happens in the first lithiation even though FFDXM start to show "defects" in the 2nd cycle. It is true that electrolyte decomposition and "early SEI" formation happen before Si lithiates around $\sim 0.1V$. While it is hard to provide the timeline/correlation for the "SEI" inhomogeneity and the Si structural change because until the formation of SEI after 1st full lithiation, it is not sure whether what formed on the electrode surface has the SEI functionality. I have reservation for the indication that SEI inhomogeneity leads to structural defects. Also, in spite of the unique in-operando study, the conclusions on SEI dual-layer structure and inhomogeneity have been reported by ex-situ AFM study of SEI in various systems.

At last, the paper's preparation needs to be further improved. For example, the minimum voltage cut off in Figure 2 and Figure 4 are not clearly labeled. Zoom-in figures are needed. The same for the current curve of the CV scan.

Some of the detail information is missing. What is the beam size? What is the defect concentration? Can the authors provide some quantitative analysis on the inhomogeneity or structural defects?

I am not sure why the authors claim the deformation is elastic (page 10). I would like to see some data support to the claim.

Have the authors measured Young's modulus of the model inorganic materials in the SEI inorganic layer? Is it consistent with what obtained in Figure 6?

Reviewer #2:

Remarks to the Author:

The paper studies a Si-based LIB system by the combination of different in operando and ex situ techniques. These include a novel, high sensitivity, high spatial resolution, in operando technique, FFDXM, providing complementary information about local structural deformation within single-crystal Si electrode. This multi-technique study sheds light on the role of SEI inhomogeneities on early-stage defect formation within the Si electrode and highlights the significance of those defects on the subsequent larger scale deformation observable by non-local techniques. Such multi-technique in

operando study is of critical importance to develop strategies to prevent early cell failure and to improve the cyclability of Si-based LIB's. As such, the results presented in the manuscript will be of interest for others and will influence this field.

I recommend the manuscript for publication with modifications. Please find my arguments below.

The major claims of the paper:

The authors include full field X-ray diffraction microscopy (FFDXM), a novel and promising new method into their study. This qualitative, in operando technique was combined with other in operando and ex-vivo techniques (e.g. operando AFM, ESM, XPS, Young modulus analysis, 3D Reciprocal Space Mapping) to obtain direct evidence for early-stage structural defect formation in single-crystal pristine Si just below the lithiated Si layer. The early-stage defects appear just after the start of the SEI formation. The paper proposes a model describing the underlying processes based on direct observations.

However, I am missing that the importance of the multi-technique approach is not highlighted in the paper. For example, mentioning only FFDXM in the abstract gives the impression that most of the important results have been obtained by this technique. I suggest the reformulation of the abstract in order to make it clear that such multi-technique approach, including in operando local methods, was crucial for this complex study.

Certain ambiguous statements should be specified or reformulated.

- For example in lines 30-31 it is claimed that "little attention has been paid to understanding the role of the SEI formed at the Si electrode surface". However, e.g. the review of Zhang et al. "Design superior solid electrolyte interface on Silicon anodes for high-performance Lithium-ion battery" *Nanoscale* 11(41), 2019, cites quite a few papers on this subject. I recommend that the authors include this reference into their paper.
- Another example (lines 38-40); the existence of a complex dual-layer SEI with a mostly inorganic inner-SEI layer and a more organic outer-SEI layer is not a new finding in itself (see the review of Zhang et al. and the papers cited therein).
- I recommend that the authors claim clearly and unambiguously the new results obtained by their study in the context of already existing results.

Certain statements are not convincing:

- It is not clear how the information obtained by the different analytical techniques were linked together for obtaining the conclusion of the paper. The interpretation of multi-technique datasets is a real challenge; for example different spatial resolutions (from some nanometres to some hundred μm 's), different sizes of the measured sample area, different information depths, and different volume- or surface-sensitivities have to be related to each other. Moreover, in operando and ex-vivo methods do not measure the same sample surface. The authors should include a discussion to describe how they handled this challenge for obtaining a reliable multi-technique data-set.
- Lines 375-377: I find that this statement is not convincing and not well supported by the provided data. The paper includes no measurements on the Young modulus and lithium-ion mobility after the 2nd and 3rd cycles, where the early-stage structural defects appear in the Si electrode. Moreover, the XPS technique does not provide information on the lateral compositional inhomogeneity of the measured SEI layer. All these techniques are surface sensitive compared to the high penetration depth of FFDXM. Could you please comment on the information depth of FFDXM? What thickness of the pristine Si was probed by FFDXM? It is not obvious how these information can be linked together to obtain the conclusion cited by the authors (see the remark above). Could you please comment on this?

A discussion would be useful concerning the validity of the obtained results and proposed model for other Si-based LIB systems; e.g. the constituents of the electrolyte will influence significantly the composition, thickness and morphology of the SEI. Could you please comment on the eventual presence of native SiO₂ species on the single-crystal Si electrode surface, and its effect on the SEI composition?

In general, the paper contains too much inaccurate expressions such as "little", "fast", "similar", "low", "a roughness of about..". I recommend that the authors improve this by providing quantitative information whenever possible.

Please find below some detailed remarks:

Lines 103-105: 1 frame/s cannot be considered as "fast" measurement; today 100-1000 frame/s frame-rate is available with certain 2D detectors. What is the limiting factor of the measurement speed? Do the authors foresee faster studies e.g. to understand the dynamics of the abrupt changes occurring during (de)lithiation (Figure 2, 4)?

Figure 2a-c and lines 152-164: I do not understand the interest of detailed analysis of the sum of the two individual defects (Int curves in the bottom of Fig 2a-c); the development and evolution of the individual defects during cycling is analysed in detail in the following paragraphs. I recommend shrinking or omitting this paragraph and the related subfigures, unless the author can illustrate the significant added value of this paragraph.

The 1:4.3 aspect ratios of the FFDXM images are quite confusing. I recommend that the authors correct all these images and show them with 1:1 ratio.

Figure 5:

- It is quite confusing that images 5j-k do not show the same sample dimensions (they correspond to a 5 x smaller region) as images 5a-g. Moreover, it is not clear whether the spatial resolutions of 5a-g are the same as that of 5j-k. Could you please comment on this? How these surface sensitive information obtained from small (1-10 μm x 1-10 μm) sample areas can be linked and compared to the FFDXM-based information originating from 100 μm x 430 μm areas?
- I do not see obvious "similar inhomogeneity" (line 289) between 5j and 5k, neither in the spatial distribution, nor in the relative variation of the measured values. Could you please define and quantify what you mean by "similar inhomogeneity" and provide a statistically sound quantitative comparison?
- The statement of lines 372-374 indicates that "the inhomogeneous thickness and lithium-ion mobility observed in Fig. 5j and k are characteristic for the outer-SEI layer...." This is important information, which should be included into this paragraph (lines 284-297).

Most of the images contain too much information. Moreover, most of the 2D images (e.g. 5f, 5j, 6c-g, S3) are of bad quality; the range of the represented values and the image contrast is not well adapted, some of the images seem to be unsharp (e.g. 6 f,g,j). As such, it is impossible to check the related conclusions of the authors. Just two examples; in lines 267-268, for Fig. 5b they provide "a roughness of about 2.4 nm", in lines 257-259 in Fig. S3 it is "a roughness of about 1.2 nm". Neither of these values can be "seen" from these figures. The authors should provide the appropriate mean values (roughness, thickness etc) and their statistical fluctuation instead of such approximate values.

Figure 6j is very confusing with two different X-axis, overlapping figure captions and X-axis title. Please include less information and re-organize this sub-figure.

Lines 384-385: "The density of these early stage defects was low, making them essentially invisible for non-local techniques that average over the entire sample." Could you please quantify what "low" means? Have you performed such non-local measurements? If yes, which ones and what was the analytical sensitivity?

Lines 443: "The rigidness of the thick Si substrates is necessary for high precision single-crystal diffraction measurements". Does it mean that only thick ($\sim 200 \mu\text{m}$) crystals can be studied with the method? This would strongly limit its possible application (e.g. excluding nanoparticles/thin-films?). Could you please comment on this? Could you please include a discussion on the limits of the experimental approach you used?

Reviewers' comments:

Reviewer #1 (Remarks to the Author):

Si is believed to be the next generation anode material for Li-ion batteries and fundamental understanding of the fading mechanism is essential for developing better Si anodes. With recent advances in Si structure design and performance improvement, understanding and improving SEI resiliency are currently of growing interest. From this point, the work potentially can have a broad impact. The authors in this paper used operando FFDXM, AFM and ESM to help understand the fading mechanism of Si structure and interphase evolution with cycling. It is unique capability, yet I don't find enough new knowledge beyond expectation to make it publishable in Nature Communications. Below please find my detail comments.

Comment 1: *First, I found it quite hard to understand some of the authors claims. What do the authors mean by early stage detection? The FFDXM signals start to show up in the 2nd cycle and after full lithiation. While Si has experienced expansion in the first lithiation and shrinkage in the first delithiation. What do the authors mean by “defects”, “structural defects”, what defects and defect of what?*

Comment 2: *Second, the definition of SEI to me is the solid electrolyte interphase layer on top of the anode that has the specific function. With that in mind, I tend to believe the SEI is SEI after the 1st full lithiation process. It is good to study the formation process of the SEI. Yet, I think it is more appropriate to call “pre-SEI” or “early SEI” for what detected during the first lithiation process in the operando AFM and ESM characterization.*

Comment 3: *Third, it is general belief that the SEI mostly forms in the first cycle and some in the 2nd cycle. The authors also showed support of SEI formation in 1st cycle in AFM measurement. Si structure change also happens in the first lithiation even though FFDXM start to show “defects” in the 2nd cycle. It is true that electrolyte decomposition and “early SEI” formation happen before Si lithiates around ~0.1V. While it is hard to provide the timeline/correlation for the “SEI” inhomogeneity and the Si structural change because until the formation of SEI after 1st full lithiation, it is not sure whether what formed on the electrode surface has the SEI functionality. I have reservation for the indication that SEI inhomogeneity leads to structural defects. Also, in spite of the unique in-operando study, the conclusions on SEI dual-layer structure and inhomogeneity have been reported by ex-situ AFM study of SEI in various systems.*

Comment 4: *At last, the paper's preparation needs to be further improved. For example, the minimum voltage cut off in Figure 2 and Figure 4 are not clearly labeled. Zoom-in figures are needed. The same for the current curve of the CV scan.*

Comment 5: *Some of the detail information is missing. What is the beam size? What is the defect concentration?*

Comment 6: *Can the authors provide some quantitative analysis on the inhomogeneity or structural defects?*

Comment 7: *I am not sure why the authors claim the deformation is elastic (page 10). I would like to see some data support to the claim.*

Comment 8: *Have the authors measured Young's modulus of the model inorganic materials in the SEI inorganic layer? Is it consistent with what obtained in Figure 6?*

Reviewer #2 (Remarks to the Author):

The paper studies a Si-based LIB system by the combination of different in operando and ex situ techniques. These include a novel, high sensitivity, high spatial resolution, in operando technique, FFDXM, providing complementary information about local structural deformation within single-crystal Si electrode. This multi-technique study sheds light on the role of SEI inhomogeneities on early-stage defect formation within the Si electrode and highlights the significance of those defects on the subsequent larger scale deformation observable by non-local techniques. Such multi-technique in operando study is of critical importance to develop strategies to prevent early cell failure and to improve the cyclability of Si-based LIB's. As such, the results presented in the manuscript will be of interest for others and will influence this field.

I recommend the manuscript for publication with modifications. Please find my arguments below.

The major claims of the paper:

Comment 1: *The authors include full field X-ray diffraction microscopy (FFDXM), a novel and promising new method into their study. This qualitative, in operando technique was combined with other in operando and ex-vivo techniques (e.g. operando AFM, ESM, XPS, Young modulus analysis, 3D Reciprocal Space Mapping) to obtain direct evidence for early-stage structural defect formation in single-crystal pristine Si just below the lithiated Si layer. The early-stage defects appear just after the start of the SEI formation. The paper proposes a model describing the underlying processes based on direct observations.*

However, I am missing that the importance of the multi-technique approach is not highlighted in the paper. For example, mentioning only FFDXM in the abstract gives the impression that most of the important results have been obtained by this technique. I suggest the reformulation of the abstract in order to make it clear that such multi-technique approach, including in operando local methods, was crucial for this complex study.

Comment 2: Certain ambiguous statements should be specified or reformulated.

- For example in lines 30-31 it is claimed that “little attention has been paid to understanding the role of the SEI formed at the Si electrode surface”. However, e.g. the review of Zhang et al. “Design superior solid electrolyte interface on Silicon anodes for high-performance Lithium-ion battery” *Nanoscale* 11(41), 2019, cites quite a few papers on this subject. I recommend that the authors include this reference into their paper.
- Another example (lines 38-40); the existence of a complex dual-layer SEI with a mostly inorganic inner-SEI layer and a more organic outer-SEI layer is not a new finding in itself (see the review of Zhang et al. and the papers cited therein).
- I recommend that the authors claim clearly and unambiguously the new results obtained by their study in the context of already existing results.

Comment 3: Certain statements are not convincing:

- It is not clear how the information obtained by the different analytical techniques were linked together for obtaining the conclusion of the paper. The interpretation of multi-technique datasets is a real challenge; for example different spatial resolutions (from some nanometres to some hundred μm 's), different sizes of the measured sample area, different information depths, and different volume- or surface-sensitivities have to be related to each other. Moreover, in operando and ex- vivo methods do not measure the same sample surface. The authors should include a discussion to describe how they handled this challenge for obtaining a reliable multi-technique data-set.

Comment 4: • Lines 375-377: I find that this statement is not convincing and not well supported by the provided data. The paper includes no measurements on the Young modulus and lithium-ion mobility after the 2nd and 3rd cycles, where the early-stage structural defects appear in the Si electrode. Moreover, the XPS technique does not provide information on the lateral compositional inhomogeneity of the measured SEI layer. All these techniques are surface sensitive compared to the high penetration depth of FFDXM. Could you please comment on the information depth of FFDXM? What thickness of the pristine Si was probed by FFDXM? It is not obvious how these information can be linked together to obtain the conclusion cited by the authors (see the remark above). Could you please comment on this?

Comment 5: A discussion would be useful concerning the validity of the obtained results and proposed model for other Si-based LIB systems; e.g. the constituents of the electrolyte will influence significantly the composition, thickness and morphology of the SEI. Could you

please comment on the eventual presence of native SiO₂ species on the single-crystal Si electrode surface, and its effect on the SEI composition?

Comment 6: *In general, the paper contains too much inaccurate expressions such as “little”, “fast”, “similar”, “low”, “a roughness of about..”. I recommend that the authors improve this by providing quantitative information whenever possible.*

Comment 7: *Please find below some detailed remarks:*

Lines 103-105: 1 frame/s cannot be considered as “fast” measurement; today 100-1000 frame/s frame-rate is available with certain 2D detectors. What is the limiting factor of the measurement speed? Do the authors foresee faster studies e.g. to understand the dynamics of the abrupt changes occurring during (de)lithiation (Figure 2, 4)?

Comment 8: *Figure 2a-c and lines 152-164: I do not understand the interest of detailed analysis of the sum of the two individual defects (Int curves in the bottom of Fig 2a-c); the development and evolution of the individual defects during cycling is analysed in detail in the following paragraphs. I recommend shrinking or omitting this paragraph and the related subfigures, unless the author can illustrate the significant added value of this paragraph.*

Comment 9: *The 1:4.3 aspect ratios of the FFDXM images are quite confusing. I recommend that the authors correct all these images and show them with 1:1 ratio.*

Comment 10: *Figure 5: • It is quite confusing that images 5j-k do not show the same sample dimensions (they correspond to a 5 x smaller region) as images 5a-g. Moreover, it is not clear whether the spatial resolutions of 5a-g are the same as that of 5j-k. Could you please comment on this? How these surface sensitive information obtained from small (1-10 μm x 1-10 μm) sample areas can be linked and compared to the FFDXM-based information originating from 100 μm x 430 μm areas?*

Comment 11: *• I do not see obvious “similar inhomogeneity” (line 289) between 5j and 5k, neither in the spatial distribution, nor in the relative variation of the measured values. Could you please define and quantify what you mean by “similar inhomogeneity” and provide a statistically sound quantitative comparison?*

Comment 12: *• The statement of lines 372-374 indicates that “the inhomogeneous thickness and lithium-ion mobility observed in Fig. 5j and k are characteristic for the outer-SEI layer...” This is important information, which should be included into this paragraph (lines 284-297).*

Comment 13: *Most of the images contain too much information. Moreover, most of the 2D images (e.g. 5f, 5j, 6c-g, S3) are of bad quality; the range of the represented values and the image contrast is not well adapted, some of the images seem to be unsharp (e.g. 6f,g,j). As such, it impossible to check the related conclusions of the authors. Just two examples; in lines 267-268, for Fig. 5b they provide “a roughness of about 2.4 nm”, in lines 257-259 in Fig. S3*

it is “a roughness of about 1.2 nm”. Neither of these values can be “seen” from these figures. The authors should provide the appropriate mean values (roughness, thickness etc) and their statistical fluctuation instead of such approximate values.

Comment 14: Figure 6j is very confusing with two different X-axis, overlapping figure captions and X-axis title. Please include less information and re-organize this sub-figure.

Comment 15: Lines 384-385: “The density of these early stage defects was low, making them essentially invisible for non-local techniques that average over the entire sample.” Could you please quantify what “low” means? Have you performed such non-local measurements? If yes, which ones and what was the analytical sensitivity?

Comment 16: Lines 443: “The rigidity of the thick Si substrates is necessary for high precision single-crystal diffraction measurements”. Does it mean that only thick (~200 μm) crystals can be studied with the method? This would strongly limit its possible application (e.g. excluding nanoparticles/thin-films?). Could you please comment on this? Could you please include a discussion on the limits of the experimental approach you used?

Reviewers' comments:

Reviewer #1 (Remarks to the Author):

Si is believed to be the next generation anode material for Li-ion batteries and fundamental understanding of the fading mechanism is essential for developing better Si anodes. With recent advances in Si structure design and performance improvement, understanding and improving SEI resiliency are currently of growing interest. From this point, the work potentially can have a broad impact. The authors in this paper used operando FFDXM, AFM and ESM to help understand the fading mechanism of Si structure and interphase evolution with cycling. It is unique capability, yet I don't find enough new knowledge beyond expectation to make it publishable in Nature Communications. Below please find my detail comments.

Answer: Our work unveils the impact of the SEI on the formation of early-stage defects. As will be shown in the answer to comment #1, FFDXM is one of the few techniques capable of detecting these defects, due to their weak structural deformation and low density. However, FFDXM alone does not allow us to understand the origin of the early-stage defects, which is why a series of complementary techniques (*operando* AFM, ESM and sputter-etched XPS) were performed. Through those investigations we discovered that the inhomogeneities in the SEI layer may be ultimately responsible for the formation of the early-stage defects. To further confirm this, we have performed additional FFDXM experiments on the same batch of Si samples coated by a homogeneous artificial-SEI layer ($\text{Li}_4\text{Ti}_5\text{O}_{12}\text{-Li}_3\text{PO}_4$). Indeed, no defects were ever observed under similar cycling conditions. This last result has since been added to the Supplementary Materials (new Fig. S6), and is part of the answer to comment #3.

We thank the reviewer for the valuable comments that motivated us to improve the quality of our work. We have made major revisions to the text as well as the figures, to clarify the new knowledge brought forward by this work and to highlight its potentially broad impact on the improvement of Si-based lithium-ion batteries.

Our answers to the comments are given below point by point:

Comment 1: *First, I found it quite hard to understand some of the authors claims. What do the authors mean by early stage detection? The FFDXM signals start to show up in the 2nd cycle and after full lithiation. While Si has experienced expansion in the first lithiation and shrinkage in the first delithiation. What do the authors mean by “defects”, “structural defects”, what defects and defect of what?*

Answer: First of all, we would like to point out that the single-crystal electrode never reached full lithiation during any of the first 3 cycles, due to the brief period it stayed below 100 mV (38 seconds for each cycle). The lithium injection ratio was estimated to be only 0.1-0.2% of the full capacity.

Secondly, the paper has been updated to better define the word “defects”.

Previously	Revised
When E reached its lowest value at +5 mV in the 2 nd cycle, two structural defects appeared (Fig. 2d), and they remain the only visible contrasts in the entire $100 \times 430 \mu\text{m}^2$ FoV up through cycle 3.	When E reached its lowest value at +5 mV in the 2 nd cycle, two defects appeared (Fig. 2d). These defects are structurally deformed Si and remain the only visible contrasts in the entire $100 \times 430 \mu\text{m}^2$ FoV up through cycle 3.

The nature of the “defects” was later revealed by 3D RSM: (text in the paper) *Fig. 3a is the processed 3D RSM of the same area. The defects appear to be larger because the entire deformed area is shown. The main contrast is dominated by the lattice tilt, which is found to be as large as 0.03° . The lattices were tilted inwards, indicating a smaller lattice parameter at the center of the defects, and hence a lower degree of lithiation in the defects than in the surrounding area after delithiation (Fig. 3b).*

We refer to them as early-stage defects (i.e. defects at their early stage of formation) because of their low density (<100 ppm, see answer to comment #5) and weak amplitude of deformation (Fig. 3a). In fact, these early-stage defects (observed in cycle 2 and 3) are too weak to be detectable by most other techniques. To illustrate this, we show results of rocking curves after the first 3 cycles (Author-reply-1-Fig.1). The rocking curve line profile is generated by integrating the intensity over a selected region of interest (RoI) on the 2D detector. Two RoIs were chosen, RoI1 ($25 \times 8 \mu\text{m}^2$) covers only the defective area while RoI2 ($430 \times 100 \mu\text{m}^2$) covers the entire field of view (FoV), as shown in Author-reply-1-Fig.1a.

Author-reply-1-Fig. 1. Rocking curve line profile generated by integrating intensity in RoI1 (b) and RoI2 (c) after cycle 1, 2 and 3. The definition of the RoIs is shown in (a).

The rocking curve line profile of RoI1 is different immediately after cycle 2. Higher intensity (marked by the red arrow in Author-reply-1-Fig.1b) was observed at about -0.03° off the Si (004) Bragg θ angle. The higher intensity was due to the scattering by the defects, and it was at this angle where all the *operando* dark field imaging (time scan) were carried out. Meanwhile, no visible changes were observed on the rocking curve line profile of RoI2 for the first 3 cycles, see Author-reply-1-Fig.1c. The integrated intensity over RoI2 represents what can be obtained with conventional XRD (one of the non-local techniques). While defects were already present inside RoI2 after the 2nd cycle, its integrated intensity showed little changes as it was dominated by scattering from the non-defective area. It is evident from this comparison that while conventional XRD is just as sensitive to lattice deformations as FFDXM, spatially resolved methods are required for detecting low density (early-stage)

defects. We have since added this figure and the relevant information to the Supplementary Materials (Fig. S2).

We note that the large FoV of FFDXM is also essential to the detection of early-stage defects. It would take Scanning X-ray Diffraction Microscopy (another spatially resolved X-ray diffraction technique) over 2 hours of counting time to cover an area of $100\times 100\ \mu\text{m}^2$ (at 100 nm resolution, 100 HZ count rate), and it might still miss the area containing the early-stage defects. In contrast, FFDXM covers an area of $430\times 100\ \mu\text{m}^2$ in just 0.1 to 1 s of counting time, making it possible to perform *operando* experiments on those low density (early-stage) defects.

Finally, we would like to reassure the reviewer that the early-stage defects described in the paper is not an isolated event. A surveyance scan covering $2\times 1\ \text{mm}^2$ of the Si electrode shows that at the end of the 6th cycle, about 20% of the surface was consistently covered with these defects. The formation of the early-stage defects does not seem to be dependent on the nature of the charging/discharging either. We have performed FFDXM experiments on a total of 6 samples with both cyclic voltammetry and galvanostatic cycling. Author-reply-1-Fig. 2 shows the result on another sample during galvanostatic charging and at constant voltage holding, extracted from FFDXM images. The onset and evolution of the defect density is comparable to what was observed with cyclic voltammetry (*cf.* answer to comment #5). This last figure has also been added to supplementary materials as Fig. S8.

Author-reply-1-Fig. 2. Si electrode potential (E) and percentage of the surface area covered by defects as a function of time during 1st galvanostatic charging and constant voltage holding of another sample. The result is calculated from FFDXM measurements of an area of $430\times 100\ \mu\text{m}^2$.

Comment 2: Second, the definition of SEI to me is the solid electrolyte interphase layer on top of the anode that has the specific function. With that in mind, I tend to believe the SEI is SEI after the 1st full lithiation process. It is good to study the formation process of the SEI. Yet, I think it is more appropriate to call “pre-SEI” or “early SEI” for what detected during the first lithiation process in the *operando* AFM and ESM characterization.

Answer: The reviewer is correct that the SEI formed on the anode surface works as a lithium-ion conductor, allowing lithium-ion transport during lithiation and delithiation. The inner-SEI is mainly formed during the first lithiation process. The outer-SEI is formed on top of the inner-SEI and continues to be developed in subsequent cycles. However, we prefer to retain using the terms inner- and outer-SEI as they are widely accepted in the literature [Author-reply-1-ref. 1-3].

[Author-reply-1-ref. 1] E. Peled, S. Menkin, Review-SEI: past, present and future, *J. Electrochem. Soc.* **2017**, *164*, A1703-A1719.

[Author-reply-1-ref. 2] K. Edstrom, M. Herstedt, and D. P. Abraham, A new look at the solid electrolyte interphase on graphite anodes in Li-ion batteries, *J. Power Sources* **2006**, *153*, 380-384.

[Author-reply-1-ref. 3] D. J. Li, H. Li, D. L. Danilov, L. Gao, X. Chen, Z. Zhang, J. Zhou, R.-A. Eichel, Y. Yang, P. H. L. Notten, Degradation mechanisms of $C_6/LiNi_{0.5}Mn_{0.3}Co_{0.2}O_2$ Li-ion batteries unraveled by non-destructive and post-mortem methods, *J. Power Sources* **2019**, *416*, 163-174.

Comment 3: *Third, it is general belief that the SEI mostly forms in the first cycle and some in the 2nd cycle. The authors also showed support of SEI formation in 1st cycle in AFM measurement. Si structure change also happens in the first lithiation even though FFDXM start to show “defects” in the 2nd cycle. It is true that electrolyte decomposition and “early SEI” formation happen before Si lithiates around ~0.1V. While it is hard to provide the timeline/correlation for the “SEI” inhomogeneity and the Si structural change because until the formation of SEI after 1st full lithiation, it is not sure whether what formed on the electrode surface has the SEI functionality. I have reservation for the indication that SEI inhomogeneity leads to structural defects. Also, in spite of the unique in-operando study, the conclusions on SEI dual-layer structure and inhomogeneity have been reported by ex-situ AFM study of SEI in various systems.*

Answer: We acknowledge that SEI formation on Si anodes has been investigated by others. However, most of the works focused on studies on SEI structure, mechanical properties, composition and growth model [Author-reply-1-ref. 4-6]. The uniqueness of our multimodal study lies in the fact that it bridges observations in the depth direction across the multi-level interfaces which include Si/Li_xSi (FFDXM), the inner-SEI (AFM, ESM, XPS) and the outer-SEI (AFM, ESM, XPS). The highlight of our work, as pointed out by the reviewer, is the correlation between inhomogeneities in the SEI and defects in the Si structure that only appear upon cycling. We first suspected such correlation after realizing that those defects were only observed after the initial formation of the SEI layer was formed (CV curve, Fig. 2). Quantitative analysis after cycle 3 (3D RSM result, Fig. 3) showed that the defect area consists of tilting of the Si lattice towards the center of the two defects, which is attributed to a different degree of lithiation at their center. We then confirmed with ESM that the lithium-ion mobility of both the inner- and outer-SEI (new Fig. 6e to 6h) was indeed not homogeneous, which to our knowledge has not been reported previously. The inhomogeneities in terms of thickness (less lithium-ion transport time if thinner, by AFM) and lithium-ion mobility (more conducting if higher, by ESM) led to heterogeneous degrees of lithiation, which in turn resulted in the lattice deformation as seen by FFDXM. It is through this multi-modal methodology that we were able to not only visualize the formation of these

early-stage defects, but also understand their origin systematically. The abstract, discussion and Fig. 1 have been updated to better reflect the importance of the multi-modal approach employed in this work.

Regarding the timeline/correlation, defects first appeared during the 2nd cycle in the FFDXM experiment (Fig. 2). In the meantime, local (*i.e.* inhomogeneous) hardening of the Si electrode was also observed during the 2nd cycle in the AFM experiment. The local hardening manifests itself as brighter contrast in the Young’s modulus map (black square, Author-reply-1-Fig. 3c). More importantly, FFDXM showed that after extensive cycling, the part with the most significant Si lattice deformation lies on top of the early-stage defects. Similarly, the cracking of the Si electrode observed by AFM (new Fig. 5d to 5g) is also tied to the local hardened area. The contrast of Young’s modulus maps (new Fig. 5k to 5n) in the paper has since been updated to highlight the local hardening. Author-reply-1-Fig. 3 has been added in the Supplementary Materials as Fig. S7 and this paragraph has been added to the discussion section in the manuscript.

Finally, to further prove that a correlation indeed exists between the inhomogeneities in the SEI and early-stage defects, FFDXM experiment was performed on single-crystal Si electrodes coated with homogeneous artificial SEI (200 nm of Li₃PO₄ on 20 nm of Li₄Ti₅O₁₂). No defects were observed even after repeated cycling (Author-reply-1-Fig. 4) which is attributed to the use of a more homogeneous SEI. This figure has since been added to the Supplementary Materials as Fig. S6 and the result has been added to the discussion section in the manuscript.

In the light of these modifications, we have revised the title of our manuscript to “*Impact of dual-layer solid-electrolyte interphase inhomogeneities on the early-stage defect formation in Si electrodes*”.

Additionally, the following changes were made:

Previously	Revised
subtitle of section “ Dual-layer SEI ”	“ Inhomogeneity of lithium-ion mobility in dual-layer SEI ”
In comparison, the SEI structure on the Si electrode is still relatively unknown.	Similar to what was found on carbon-based electrodes [24-27], the SEI on Si also has a dual-layer structure, composed of an inner- and outer-SEI layer [28-30].
From these operando Young’s modulus analyses it can be concluded that the SEI formed on Si also has a dual-layered structure, consisting of a very thin inner-SEI layer and an outer-SEI layer, which continues to grow with cycling.	Deleted to emphasize on only the new results.
The SEI formed on Si electrodes was found to be composed of a dual-layer structure. The inner-SEI layer, formed during the initial stages of lithiation, is hard and mainly composed of inorganic lithium-salt, such as LiF. It passivates the electrochemically active surface as revealed	Deleted to emphasize on only the new results.

by the suppressed current peak in the CV plots in the subsequent cycles compared to that in the first cycle.

Unlike the inner-SEI layer, the outer-SEI layer is mainly composed of softer organic lithium salts, which continues to grow with cycling.

Deleted to emphasize on only the new results.

Author-reply-1-Fig. 3. Topography and corresponding Young's modulus mapping of the single-crystal Si electrode during the 2nd (b and c) and 3rd CV-cycle (e and f). The corresponding CV range for each image is indicated by the colored curve for the 2nd (a) and 3rd (d) cycle, respectively.

Author-reply-1-Fig. 4. (a) Potential and current plot for the artificial-SEI coated single-crystal Si electrode. FFDXM image before (b) and after (c) the cycling showed no defects on the Si electrode. The artificial SEI of $\text{Li}_4\text{Ti}_5\text{O}_{12}$ (20 nm)- Li_3PO_4 (200 nm) was deposited by magnetron sputtering.

[Author-reply-1-ref. 4] Y. Zhang, N. Du, D. Yang, Design superior solid electrolyte interface on silicon anodes for high-performance lithium-ion battery, *Nanoscale*, **2019**, *11*, 19086-19104.

[Author-reply-1-ref. 5] J. Zheng, H. Zheng, R. Wang, L. Ben, W. Liu, L. Chen, L. Chen, H. Li, 3D visualization of inhomogeneous multi-layered structure and Young's modulus of the solid electrolyte interphase (SEI) on silicon anodes for lithium ion batteries, *Phys. Chem. Chem. Phys.* **2014**, *16*, 13229-13238

[Author-reply-1-ref. 6] A. Wang, S. Kadam, H. Li, S. Shi, Y. Qi, Review on modeling of anode solid electrolyte interphase (SEI) for lithium-ion batteries. *Npj Comput. Mater.* **2018**, *4*, 15.

Comment 4: At last, the paper's preparation needs to be further improved. For example, the minimum voltage cut off in Figure 2 and Figure 4 are not clearly labeled. Zoom-in figures are needed. The same for the current curve of the CV scan.

Answer: We thank the reviewer for this comment. The following changes were made:

The voltage cut-off has been labeled and CV plots have been magnified in Fig. 2 and Fig. 4.

Per comment of the other reviewer, the total integrated intensity curves of cycle 1 to 3 have been moved to the Supplementary Materials as new Fig. S1.

Higher resolution FFDXM images were used to replace the old ones in Fig. 2e.

The layout of Fig. 3 has been updated. The reference color wheel (previously Fig. 3c) has been moved to the inset of Fig. 3a. The FFDXM raw image (previously Fig. 3a) has been removed as it essentially contains the same information as Fig. 2e10.

To further improve the paper presentation, the result of the Young's modulus measurement in Fig. 6 was moved to Fig. 5. Additionally, their contrasts have been updated to highlight the local hardening of the sample as described in the answer to comment #3. Result on the lithium-ion mobility (previously Fig. S5 and Fig. 5j) has been moved to Fig. 6, as described in the answer to comment #6.

Comment 5: *Some of the detail information is missing. What is the beam size? What is the defect concentration?*

Answer: We thank the reviewer for this comment. Additional information has been added to the main and methods section:

Added citation for the fabrication of the objective lenses used in this study to the *Methods* section: A set of CRLs (SU-8, made by the Karlsruhe Institute of Technology, 6 μm apex radius of curvature [Author-reply-1-ref. 7]) was mounted behind the sample to image the diffracted beam.

Added information about the illumination or beam size to the *Methods* section: The illuminated sample area is $980 \text{ (H)} \times 240 \text{ (V)} \mu\text{m}^2$. The pre-focusing by the in vacuum Beryllium transfocator at 62 meters upstream of the sample allows a high incoming photon flux to be achieved with a near parallel beam (convergence $< 0.03 \text{ mrad}$).

Added information on the final surface area covered by the defects to the description under the new Fig. S3 of the Supplementary Materials: A surveyance scan covering $2 \times 1 \text{ mm}^2$ of the Si electrode shows that at the end of the 6th cycle, about 20% of the surface was consistently covered with these defects.

Added information on the evolution of the surface area covered by the defects for cycle 1 to 6 (Author-reply-1-Fig. 5) to the Supplementary Materials as the new Fig. S3. The result echoes what was mentioned in the answer to comment #1. The early-stage defects covered a mere 0.3% of the entire surface area ($< 100 \text{ ppm}$ of the volume considering an X-ray penetration depth of $200 \mu\text{m}$) and would hence be undetectable to non-local techniques such as conventional XRD. The defect coverage only became significant (rose to $\sim 20\%$) after 2000 seconds of accumulated holding at 5 mV, which is comparable with what was shown in Author-reply-1-Fig. 2 for a different sample.

Author-reply-1-Fig. 5. Potential, current and defect coverage plot for cycle 1-6.

[Author-reply-1-ref. 7] F. Marschall, A. Last, M. Simon, H. Vogt, J. Mohr, Simulation of aperture-optimised refractive lenses for hard X-ray full field microscopy, *Opt. Express* **2016**, *24*, 10880-10889.

Comment 6: Can the authors provide some quantitative analysis on the inhomogeneity or structural defects?

Answer: We have added to the manuscript the following quantitative descriptions of the inhomogeneities:

- (1) The RMS roughness of the inner-SEI which is 2.4 nm.
- (2) The measured amplitude in ESM experiment is proportional to the lithium-ion mobility. The histogram of ESM amplitudes of both outer- and inner-SEI has been added to the manuscript as the new Fig. 6f and 6h, respectively. For the reviewer's convenience, we have also attached the histograms here, as shown Author-reply-1-Fig. 6. The histograms also offer comparison between the mean lithium ion mobility in the inner- (b) and in the outer-SEI (a) layer. We found that the lithium-ion mobility of outer-SEI (a) is ~ 1.5 times the value of the inner-SEI (b), which is explained by a higher lithium-ion conductivity for Li_2CO_3 and ROCO_2Li (main components of the outer-SEI) than for LiF (main component of the inner-SEI) [Author-reply-1-ref. 8-11].

To our knowledge, this is new information to the scientific community. Its implication was discussed in the discussion section of the manuscript.

Quantitative analysis of the lattice deformation in the defect area has already been shown in Fig. 3a. Additionally, we have added the evolution of the surface area covered by the defects for cycle 1 to 6 (Author-reply-1-Fig. 5) to Supplementary Materials (Fig. S3), as discussed in the reply to comment #5.

Author-reply-1-Fig. 6. ESM amplitude histogram for outer- (a) and inner-SEI (b), which has been added to the revised manuscript as the new Fig. 6f and 6h.

[Author-reply-1-ref. 8] S. Shi, Y. Qi, H. Li, L. G. H. Jr., Defect thermodynamics and diffusion mechanisms in Li_2CO_3 and implications for the solid electrolyte interphase in Li-ion batteries, *J. Phys. Chem. C* **2013**, *117*, 8579-8593.

[Author-reply-1-ref. 9] O. Borodin, G. V. Zhang, P. N. Ross, K. Xu, Molecular dynamics simulations and experimental study of lithium ion transport in dilithium ethylene dicarbonate, *J. Phys. Chem. C* **2013**, *117*, 7433-7444.

[Author-reply-1-ref. 10] D. Bedrov, O. Borodin, J. B. Hooper, Li^+ transport and mechanical properties of model solid electrolyte interphases (SEI): insight from atomistic molecular dynamics simulations, *J. Phys. Chem. C* **2017**, *121*, 16098-16109.

[Author-reply-1-ref. 11] J. Pan, Y-T. Cheng, Y. Qi, General method to predict voltage-dependent ionic conduction in a solid electrolyte coating on electrodes, *Phys. Rev. B* **2015**, *91*, 134116.

Comment 7: *I am not sure why the authors claim the deformation is elastic (page 10). I would like to see some data support to the claim.*

Answer: We claimed that the deformation was mainly elastic because the FFDXM contrast became much weaker just before the reversal of the sign of the applied stress (*i.e.* when the applied force was close to 0). This for instance was observed for defect Area I when it first appeared during the 2nd delithiation (Fig. 2e3) and more evidently during the 3rd lithiation (Fig. 2e7) and delithiation (Fig. 2e9). This also holds true for the new defects that appeared during and disappeared at the end of the 4th cycle (red circles in Author-reply-1-Fig. 6.). Even though the off-Bragg time scans do not provide quantitative information on the defects, the measured intensity (FFDXM contrast) is roughly proportional to the volume of the defects. An almost disappearance of the contrast (Fig. 2e7) would imply that most of the deformed Si was restored to normal which can only happen if the deformation was mainly elastic.

The paper has been updated to clarify this claim: (1) The much weaker contrast in the corresponding FFDXM image (Fig. 2e7) indicated that the early-stage defects were mainly elastic, as most of the deformed Si was, at this point, restored to normal. (2) These newly formed defects were, at this point, mainly elastic as confirmed by the vanished contrast after delithiation (Fig. 4d13).

We would like to point out, however, the above statement remained true only for the early-stage defects. Upon repeated cycling, plasticity became dominating in the defect area, which is consistent with other observations in the literature [Author-reply-1-ref. 12].

Author-reply-1-Fig. 6. Same as Fig. 4d11, 4d12, 4d13, the newly appeared defects are highlighted by red circles.

[Author-reply-1-ref. 12] M. Pharr, Z. G. Suo, J. J. Vlassak, Measurements of the fracture energy of lithiated silicon electrodes of Li-ions batteries, *Nano Lett.* **2013**, *13*, 5570-5577.

Comment 8: Have the authors measured Young's modulus of the model inorganic materials in the SEI inorganic layer? Is it consistent with what obtained in Figure 6?

Answer: Before the *operando* AFM experiment in PF-QNM mode, tip calibration procedure [Author-reply-1-ref. 13] was rigorously followed to ensure reproducible and quantitative Young's modulus measurements. In the present work, special care has been taken to calibrate the tip with reference silica and highly oriented pyrolytic graphite (HOPG) samples. The measured Young's modulus of the presently used Si wafer surface (under dry condition) matches well with the reference sample and the literature. It is worth to note that crystalline conditions, surface parameters and thickness of samples will affect the measured Young's

modulus data. The SEI, both inner- and outer-, are mixtures composed of inorganic and organic materials. This adds difficulties to the extraction of absolute data for a pure material, e.g. LiF. Measurement in liquid on slightly or partially porous materials such as SEI will also affect the result when comparing to that on a dry surface. Therefore, comparison against pure model systems under different conditions has only limited significance and doing so often leads to misinterpretations of the data, which is why it is not presented in the paper. Nonetheless, the average Young's modulus measured on the inner-SEI, which contains mainly inorganic LiF and Li₂O, is around 5 GPa (new Fig. 5i). The value is lower than the parameter of pure LiF and Li₂O but is in excellent agreement with previously reported work for the SEI, which varies between 0.4 to 4.5 GPa [Author-reply-1-ref. 5, 14].

[Author-reply-1-ref.13] https://www.bruker.com/fileadmin/user_upload/8-PDF-Docs/SurfaceAnalysis/AFM/ApplicationNotes/AN128-RevB0-Quantitative_Mechanical_Property_Mapping_at_the_Nanoscale_with_PeakForceQNM-AppNote.pdf

[Author-reply-1-ref. 14] H. Shin, J. Park, S. Han, A. M. Sastry, W. Lu, Component-/structure-dependent elasticity of solid electrolyte interphase layer in Li-ion batteries: Experimental and computational studies, *J. Power Sources* **2015**, 277, 169-179.

Reviewer #2 (Remarks to the Author):

The paper studies a Si-based LIB system by the combination of different in operando and ex situ techniques. These include a novel, high sensitivity, high spatial resolution, in operando technique, FFDXM, providing complementary information about local structural deformation within single-crystal Si electrode. This multi-technique study sheds light on the role of SEI inhomogeneities on early-stage defect formation within the Si electrode and highlights the significance of those defects on the subsequent larger scale deformation observable by non-local techniques. Such multi-technique in operando study is of critical importance to develop strategies to prevent early cell failure and to improve the cyclability of Si-based LIB's. As such, the results presented in the manuscript will be of interest for others and will influence this field.

I recommend the manuscript for publication with modifications. Please find my arguments below.

Answer: We thank the reviewer very much for the positive feedback and his recommendation for publication in Nature Communications. Our answers to the comments are given below point by point.

The major claims of the paper:

Comment 1: *The authors include full field X-ray diffraction microscopy (FFDXM), a novel and promising new method into their study. This qualitative, in operando technique was combined with other in operando and ex-vivo techniques (e.g. operando AFM, ESM, XPS, Young modulus analysis, 3D Reciprocal Space Mapping) to obtain direct evidence for early-stage structural defect formation in single-crystal pristine Si just below the lithiated Si layer. The early-stage defects appear just after the start of the SEI formation. The paper proposes a model describing the underlying processes based on direct observations.*

However, I am missing that the importance of the multi-technique approach is not highlighted in the paper. For example, mentioning only FFDXM in the abstract gives the impression that most of the important results have been obtained by this technique. I suggest the reformulation of the abstract in order to make it clear that such multi-technique approach, including in operando local methods, was crucial for this complex study.

Answer: We agree with the reviewer that the importance of the multi-modal techniques with needs to be highlighted. We have restructured the abstract and the discussion accordingly. Additionally, Fig. 1 and its caption have also been updated to better reflect the importance of the multi-modal approach in this work.

Comment 2: *Certain ambiguous statements should be specified or reformulated.*

- *For example in lines 30-31 it is claimed that “little attention has been paid to understanding the role of the SEI formed at the Si electrode surface”. However, e.g. the review of Zhang et al. “Design superior solid electrolyte interface on Silicon anodes for high-performance Lithium-ion battery” Nanoscale 11(41), 2019, cites quite a few papers on this subject. I recommend that the authors include this reference into their paper.*
- *Another example (lines 38-40); the existence of a complex dual-layer SEI with a mostly inorganic inner-SEI layer and a more organic outer-SEI layer is not a new finding in itself (see the review of Zhang et al. and the papers cited therein).*
- *I recommend that the authors claim clearly and unambiguously the new results obtained by their study in the context of already existing results.*

Answer: We thank the reviewer for this comment. We acknowledge that SEI formation on Si anodes has been investigated by others. However, most of the works focused on studies on SEI structure, mechanical properties, composition and growth model [Author-reply-2-ref. 1-3]. The highlight of our work is the correlation between inhomogeneities in the SEI and defects in the Si structure that only appear upon cycling. To clarify this, we have changed “*little attention has been paid to understanding the role of the SEI formed at the Si electrode surface*” to “*little attention has been paid to understanding its role in the mechanical failures of the electrodes.*” in the abstract.

The corresponding citations [Author-reply-2-ref. 1-3] have since been added to the paper as new references [28-30].

Additionally, the following changes were made:

Previously	Revised
subtitle of section “Dual-layer SEI”	“ Inhomogeneity of lithium-ion mobility in dual-layer SEI ”
In comparison, the SEI structure on the Si electrode is still relatively unknown.	Similar to what was found on carbon-based electrodes [24-27], the SEI on Si also has a dual-layer structure, composed of an inner- and outer-SEI layer [28-30].
From these operando Young’s modulus analyses it can be concluded that the SEI formed on Si also has a dual-layered structure, consisting of a very thin inner-SEI layer and an outer-SEI layer, which continues to grow with cycling.	Deleted to emphasize on only the new results.
The SEI formed on Si electrodes was found to be composed of a dual-layer structure. The inner-SEI layer, formed during the initial stages of lithiation, is hard and mainly composed of inorganic lithium-salt, such as LiF. It passivates the electrochemically active surface as revealed by the suppressed current peak in the CV plots in the subsequent cycles compared to that in the first cycle.	Deleted to emphasize on only the new results.
Unlike the inner-SEI layer, the outer-SEI layer is mainly composed of softer organic lithium salts, which continues to grow with cycling.	Deleted to emphasize on only the new results.

[Author-reply-2-ref. 1] Y. Zhang, N. Du, D. Yang, Design superior solid electrolyte interface on silicon anodes for high-performance lithium-ion battery, *Nanoscale*, **2019**, *11*, 19086-19104.

[Author-reply-2-ref. 2] J. Zheng, H. Zheng, R. Wang, L. Ben, W. Liu, L. Chen, L. Chen, H. Li, 3D visualization of inhomogeneous multi-layered structure and Young’s modulus of the solid electrolyte interphase (SEI) on silicon anodes for lithium ion batteries, *Phys. Chem. Chem. Phys.* **2014**, *16*, 13229-13238

[Author-reply-2-ref. 3] A. Wang, S. Kadam, H. Li, S. Shi, Y. Qi, Review on modeling of anode solid electrolyte interphase (SEI) for lithium-ion batteries. *Npj Comput. Mater.* **2018**, *4*, 15.

Comment 3: *Certain statements are not convincing:*

- *It is not clear how the information obtained by the different analytical techniques were linked together for obtaining the conclusion of the paper. The interpretation of multi-technique datasets is a real challenge; for example different spatial resolutions (from some nanometres to some hundred μm 's), different sizes of the measured sample area, different information depths, and different volume- or surface-sensitivities have to be related to each other. Moreover, in operando and ex- vivo methods do not measure the same sample surface.*

The authors should include a discussion to describe how they handled this challenge for obtaining a reliable multi-technique data-set.

Answer: Our work unveils the impact of the SEI on the formation of early-stage defects. As will be shown in the answer to comment #7 and #15, FFDXM is one of the few techniques capable of detecting these defects, due to their weak lattice deformation and low density. However, FFDXM alone does not allow us to understand the origin of the early-stage defects, which is why a series of complementary techniques (*operando* AFM, ESM and sputter-etched XPS) were performed. Through those investigations we discovered that the inhomogeneities in the SEI layer may be ultimately responsible for the formation of the structural defects. To further confirm this, we have performed additional FFDXM experiments on the same batch of Si samples coated by a homogeneous artificial-SEI layer ($\text{Li}_4\text{Ti}_5\text{O}_{12}\text{-Li}_3\text{PO}_4$). Indeed, no defects were ever observed under similar cycling conditions. This result is discussed in detail in the answer to comment #4. We agree with the reviewer that the link between the results obtained from different techniques was not sufficiently clarified in the text. A new paragraph has been added to the discussion of the paper to specifically address this problem.

The mismatch in spatial resolution and field of view (FoV) between the FFDXM and AFM is in fact welcomed. As stated in the paper, the deformation in the lattice often extends much further than the physical size of the defects. The 100 nm spatial resolution of FFDXM is adequate for the investigation of the nature of the defects because their lattice deformation has the size of roughly $4 \times 4 \mu\text{m}^2$ (RSM result, Fig. 3). Its large FoV allowed us to detect those sparsely distributed defects when they were initially formed. AFM, on the other hand, measures the physical size of the defects. Its finer spatial resolution (20 nm) is required to resolve cracks ($1 \times 1 \mu\text{m}^2$) and topographic inhomogeneities ($0.1 \times 0.1 \mu\text{m}^2$) as shown in Fig. 5. This paragraph has been added to the discussion section of the manuscript. The mismatch in information depth / surface sensitivity will be discussed in the answer to comment #4.

Regarding the reliability of the dataset: We ensure the reliability by using, across the different characterization platforms, the same batch of samples and type/mode of cycling equipment. Moreover, to ascertain that the early-stage defects were reproducible and not just an isolated event, we have performed FFDXM measurements on a total of 6 samples (the one presented in the paper was the last of the 6 samples), using different cycling parameters and procedures (cyclic voltammetry, galvanostatic). Each time, we were able to observe the onset of those defects as well as their evolution upon cycling. We have included the evolution of the surface coverage by the defects during cycle 1-6 in the answer to comment #15 and in the supplementary materials (Fig. S3). The result is comparable to what was observed on another sample during galvanostatic charging and constant voltage holding (Fig. S8, *cf.* the answer to comment #15).

Comment 4: • *Lines 375-377: I find that this statement is not convincing and not well supported by the provided data. The paper includes no measurements on the Young modulus and lithium-ion mobility after the 2nd and 3rd cycles, where the early-stage structural defects appear in the Si electrode. Moreover, the XPS technique does not provide information on the lateral compositional inhomogeneity of the measured SEI layer. All these techniques are surface sensitive compared to the high penetration depth of FFDXM. Could you please comment on the information depth of FFDXM? What thickness of the pristine Si was probed*

by FFDXM? It is not obvious how these information can be linked together to obtain the conclusion cited by the authors (see the remark above). Could you please comment on this?

Answer: The penetration depth of X-ray at 19.7 keV for the (004) Si Bragg reflection is roughly 200 μm , which is the thickness of pristine Si being probed by FFDXM. However, because pristine Si diffracts only at a very narrow angular range (given by the Darwin's width, the energy resolution and the convergence angle), they were not "seen" when the sample is slightly rotated off the Bragg condition. This is further illustrated in Fig. S9b. At -0.03° off the Bragg angle, the main contributors to the scattered photons are thermal diffuse scattering and scattering by deformed Si. As a result, our FFDXM measurement is extremely sensitive to the defects (which happened to be at the Si/Li_xSi interface) and is not affected by the thickness of the probed pristine Si. This above is the reason why we can link the information acquired by FFDXM with those acquired by AFM/ESM. It is worth mentioning that if the imaging was performed for instance exactly at the Bragg angle, the defects would not have been visible as they would be buried in the much stronger scattering of pristine Si.

We thank the reviewer for pointing out that without measurements on the Young's modulus after the 2nd and 3rd cycle, the timeline correlation between FFDXM and AFM is not well established. This missing information (Author-reply-2-Fig. 2) has been added to the Supplementary materials (new Fig. S7). More specifically, *operando* FFDXM showed that the early stage defects first appeared during the 2nd cycle (Fig. 2). In the meantime, local (*i.e.* inhomogeneous) hardening of the Si electrode was also observed during the 2nd cycle in the AFM experiment. The local hardening manifests itself as brighter contrast in the Young's modulus map (black square, Author-reply-2-Fig. 2c). More importantly, FFDXM showed that after extensive cycling, the part with the most significant Si lattice deformation lies on top of the early-stage defects. Similarly, the cracking of the Si electrode observed by AFM (new Fig. 5e to 5g) is also tied to the local hardened area. The contrast of Young's modulus maps (new Fig. 5k to 5n) in the paper has since been updated to highlight the local hardening observed in the Young's modulus map. This paragraph has been added to the discussion section in the manuscript.

Finally, to further prove that a correlation indeed exists between the inhomogeneities in the SEI and early-stage defects, FFDXM experiments were performed on single-crystal Si electrodes coated with homogeneous artificial SEI layers (200 nm of Li₃PO₄ on 20 nm of Li₄Ti₅O₁₂). No defects were observed even after repeated cycling (Author-reply-2-Fig. 3) which is attributed to the use of a homogeneous SEI. This figure has since been added to the Supplementary Materials as the new Fig. S6 and the result has been added to the discussion section in the manuscript.

In the light of these modifications, we have revised the title of our manuscript to "*Impact of dual-layer solid-electrolyte interphase inhomogeneities on the early-stage defect formation in Si electrodes*"

Author-reply-2-Fig. 2. Topography and corresponding Young's modulus mapping of the single-crystal Si electrode during the 2nd (b and c) and 3rd CV-cycle (e and f). The corresponding CV range for each image is indicated by the colored curve for the 2nd (a) and 3rd (d) cycle, respectively.

Author-reply-2-Fig. 3. (a) Potential and current plot for the artificial-SEI coated single-crystal Si electrode. FFDXM image before (b) and after (c) the cycling showed no defects on the Si electrode. The artificial SEI of $\text{Li}_4\text{Ti}_5\text{O}_{12}$ (20 nm)- Li_3PO_4 (200 nm) was deposited by sputtering-magnetron deposition method.

Comment 5: A discussion would be useful concerning the validity of the obtained results and proposed model for other Si-based LIB systems; e.g. the constituents of the electrolyte will influence significantly the composition, thickness and morphology of the SEI. Could you please comment on the eventual presence of native SiO₂ species on the single-crystal Si electrode surface, and its effect on the SEI composition?

Answer: We agree with the reviewer that the constituents of the electrolyte, e. g. lithium-salt chemistry, concentration, solvent composition and additives, will significantly affect the SEI composition, thickness and morphology. Additionally, the electrochemical conditions, such as voltage scanning rate, current rate and resting time, also play a role in the SEI formation. Our goal is to highlight the effect of inhomogeneous SEI on the structural deformation in the electrode. To do so, we used single-crystal Si wafers. The Si near the surface participates in the (de)lithiation while the Si right below serves as a sensor for structural deformations. After decades of perfection, Si wafers to date are among the best quality man-made crystals. It has very low surface roughness, which guarantees a homogeneous starting point for the SEI formation. Its near perfect crystallinity makes it possible for us to detect any early-stage defects with weak lattice distortion and low density. For comparison, thin film or nanoparticle Si has been used as electrode in the literature. In those studies, however, it is difficult to detect weak mechanical deformation as the electrode is either amorphous or of poor crystalline quality. To put it differently, our use of bulk Si as a sensor to structural deformation allowed us to visualize early-stage defects that would otherwise be difficult to detect. We further correlate the formation of these defects to the inhomogeneities in the SEI layer and show (cf. answer to comment #4) that these defects are absent under the homogeneous artificial SEI (Li₄Ti₅O₁₂-Li₃PO₄). This correlation between the SEI inhomogeneities and structural deformation may have more profound implications. For instance, it may explain the unexpected yet consistent crack observations in extremely thin (20 nm) Si film electrodes, despite a theoretical critical thickness of 100 nm. More generally, we hope to accentuate the importance of minimizing the inhomogeneities in the SEI (natural or artificial) on improving the structural stability of Si-based LIB. The paper has been updated to reflect the above discussion.

We thank the reviewer for pointing out the presence of native SiO₂ at the surface. In our case, we intentionally kept the thin native SiO₂ (i.e. unetched), for two reasons. First, it has been previously demonstrated that, under identical electrochemical conditions, the SEI formed on etched Si has a higher roughness compared to those formed on native SiO₂ [Author-reply-2-ref. 4]. In this respect, the presence of native SiO₂ will lead to a less rough SEI, which is good for the observation of the defects in their early stage. Secondly and more importantly, all Si samples are inevitably covered with native oxide in real batteries. Performing measurements on native SiO₂ covered Si electrodes thus allowed us to conduct our experiments closer to the real-life scenario. The effect of native SiO₂ species on the SEI composition has been studied elsewhere [Author-reply-2-ref. 4, 5]. It was shown that the presence of native SiO₂ shifts the electrolyte reduction from 1.6 V to a lower potential, due to the insulating SiO₂ kinetically hindering the electrolyte reduction. This shifting of the SEI formation potential was also observed in our study.

The implication of having native SiO₂ has been added to the discussion section of the manuscript: *Optimizing the thickness and roughness of the inner-SEI is thus more rewarding than tuning those of the outer one. The properties of the inner-SEI are known to be affected by*

the surface of the electrode, due to their direct contact. As is the case for this study, the presence of native silicon oxide promotes the formation of a less rough and thinner SEI [43, 44]. The use of rigid Si wafer with atomically flat surface also helps reduce the roughness of the SEI layer, which is favorable for observing the defects in their early stage.

[Author-reply-2-ref. 4] K. W. Schroder, A. G. Dylla, S. J. Harris, L. J. Webb, K. J. Stevenson, Role of surface oxides in the formation of solid-electrolyte interphases at silicon electrodes for lithium-ion batteries, *ACS Appl. Mater. Interfaces* **2014**, 6, 21510-21524.

[Author-reply-2-ref. 5] C. Cao, I. I. Abate, E. Sivonxay, B. Shyam, C. Jia, B. Moritz, T. P. Devereaux, K. A. Persson, H-G. Steinrück, M. F. Toney, Solid electrolyte interphase on native oxide-terminated silicon anodes for Li-ion batteries, *Joule*, **2019**, 3, 762-781.

Comment 6: *In general, the paper contains too much inaccurate expressions such as “little”, “fast”, “similar”, “low”, “a roughness of about..”. I recommend that the authors improve this by providing quantitative information whenever possible.*

Answer: We thank the reviewer for pointing out the use of many inaccurate expressions. We have carefully checked the whole manuscript and replaced the relevant statements with more precise descriptions (highlighted in red).

Comment 7: *Please find below some detailed remarks:*

Lines 103-105: 1 frame/s cannot be considered as “fast” measurement; today 100-1000 frame/s frame-rate is available with certain 2D detectors. What is the limiting factor of the measurement speed? Do the authors foresee faster studies e.g. to understand the dynamics of the abrupt changes occurring during (de)lithiation (Figure 2, 4)?

Answer: The reviewer is correct in that most time-resolved X-ray microscopy studies today operates at kHz if not MHz range. Those microscopes image transmitted photons with an incident flux of 10^6 photon/s/ μm^2 for monochromatic X-ray, and many more if a pink/white beam is used. FFDXM images diffracted photons of monochromatic X-ray. For a perfect crystal, such as Si and a strong reflection at (004), the diffracted photon flux is only slightly weaker than the transmitted one. However, at off-Bragg conditions, where most of the experiment was carried out, the scattered photon is roughly 4 orders of magnitude weaker, as shown in Fig. S9b. The efficiency of the objective lenses adds another factor of 0.1. It is through those factors (monochromatic vs polychromatic, off-Bragg, lens efficiency) that a longer counting time is required in the case of FFDXM. It is our sincere belief that the current frame rate is adequate for this study. The sharp rise in integrated intensity during the 2nd lithiation was sufficiently described by 7 data points (*i.e.* there are 7 FFDXM images between point 1 and point 2 in Fig. 2b), which is why a further increase of the frame rate may not be warranted.

We also note that as a full field imaging method, FFDXM is considerably faster than its scanning counterpart. It would take Scanning X-ray Diffraction Microscopy (another spatially resolved X-ray diffraction technique) over 2 hours of counting time to cover an area of $100 \times 100 \mu\text{m}^2$ (at 100 nm resolution, 100 Hz count rate), and it might still miss the area

containing the early-stage defects. In contrast, FFDXM covers an area of $430 \times 100 \mu\text{m}^2$ in just 0.1 to 1 s of counting time, making it possible to perform *operando* experiments on those low density (early-stage) defects.

Comment 8: Figure 2a-c and lines 152-164: I do not understand the interest of detailed analysis of the sum of the two individual defects (Int curves in the bottom of Fig 2a-c); the development and evolution of the individual defects during cycling is analysed in detail in the following paragraphs. I recommend shrinking or omitting this paragraph and the related subfigures, unless the author can illustrate the significant added value of this paragraph.

Answer: We agree with the reviewer that the heterogeneous evolution of individual defect is more interesting to be discussed in the main text. We have restructured Fig. 2 and moved the total Int curves of the first three cycles to the Supplementary Materials as the new Fig. S1. The relevant discussion about the total Int curves was also moved to the Supplementary Materials. For the reviewer's information, we have attached the new Fig. 2 as Author-Reply-2-Fig. 4.

Author-reply-2-Fig. 4. the new version of Fig. 2 without the total Int. curve.

Comment 9: *The 1:4.3 aspect ratios of the FFDXM images are quite confusing. I recommend that the authors correct all these images and show them with 1:1 ratio.*

Answer: We understand that the 1:4.3 aspect ratio of the FFDXM images can be a bit confusing to the readers. However, we would like to ask the reviewer's permission to keep it because they are the as-seen (raw) images on the 2D detector, due to projection of the diffracted beam at shallow exit angles (13.4°). Moreover, stretching it back to 1:1 makes the images odd-looking esthetically. To avoid confusion, we have carefully positioned scale bars and explanations throughout the paper.

Comment 10: *Figure 5: • It is quite confusing that images 5j-k do not show the same sample dimensions (they correspond to a 5 x smaller region) as images 5a-g. Moreover, it is not clear whether the spatial resolutions of 5a-g are the same as that of 5j-k. Could you please comment on this? How these surface sensitive information obtained from small (1-10 μm x 1-10 μm) sample areas can be linked and compared to the FFDXM-based information originating from 100 μm x 430 μm areas?*

Answer: We thank the reviewer for this excellent comment.

As stated in the answer to comment #3, the mismatch in spatial resolution and in FoV between the FFDXM and the AFM is in fact welcomed. FFDXM is sensitive to lattice distortions which often extend further in space than the physical size of the defects as measured by the AFM. Moreover, the two techniques offer rather complementary views to the same problem, which are the inhomogeneities near the surface of the electrode (for surface sensitivity of FFDXM, please refer to the answer to comment #4). FFDXM is highly sensitive to deformation in the Si matrix but is blind to anything that is happening in the SEI. AFM (ESM) on the other hand measures essentially the SEI and is insensitive to weak changes in the underlying Si.

Regarding the differences between Fig. 6e (previously Fig. 5k) and Fig. 5a-g, the ESM measurements were indeed carried out on an area that is 5 times smaller, but with a 5 times higher spatial resolution (all the present AFM and ESM images are recorded with a pixel density of 256 data points per line). The smaller area was chosen to ensure a stable resonance signal for quantitative ESM mapping.

Comment 11: *• I do not see obvious “similar inhomogeneity” (line 289) between 5j and 5k, neither in the spatial distribution, nor in the relative variation of the measured values. Could you please define and quantify what you mean by “similar inhomogeneity” and provide a statistically sound quantitative comparison?*

Answer: We thank the reviewer for this comment. We agree it is misleading to use the term “similar”. Indeed, the ESM amplitude and topography mapping are not connected despite being acquired simultaneously. We have deleted the word “similar” in the relevant sentence.

Additionally, to improve the paper's presentation as per the request of the other reviewer, result of the Young's modulus measurement (previously Fig. 6) was moved to Fig. 5. Result on the lithium-ion mobility (previously Fig. S5b and Fig. 5k) has been moved to Fig. 6. The

corresponding topographic maps (previously Fig. S5a and Fig. 5j) were removed and replaced by histograms of the ESM amplitude.

Comment 12: • *The statement of lines 372-374 indicates that “the inhomogeneous thickness and lithium-ion mobility observed in Fig. 5j and k are characteristic for the outer-SEI layer....” This is important information, which should be included into this paragraph (lines 284-297).*

Answer: We thank the reviewer for this excellent comment. We agree with the reviewer and have since moved the sentence containing “the inhomogeneous thickness and lithium-ion mobility observed in Fig. 5j and k are characteristic for the outer-SEI layer....” to the paragraph “Result of 3D RSM (Fig. 3) indicated heterogeneous lithiation...”. Additionally, the discussion section of the manuscript has been updated to highlight its implications:

(1) This heterogeneous lithiation is likely to be caused by two major factors: inhomogeneities in the thickness of the lithium-ion conducting SEI layer (longer transportation time for thicker regions) and in the mobility of the lithium-ion (less conducting for lower mobility). Both were supported by the present study. On one hand, operando topographic measurement showed a 2-fold increase in RMS roughness (standard deviation of layer thickness) in the initially formed SEI (Fig. 5b). On the other hand, the ex situ ESM (Fig. 6e) result revealed a strong variation in lithium-ion mobility of the SEI layer, with some areas of the sample conducting lithium-ion more than 4 times less than other areas. (2) We note that the average lithium-ion mobility of the outer-SEI layer (Fig. 6f) is ~1.5 times that of the inner one (Fig. 6h), which is explained by a higher lithium-ion conductivity for Li_2CO_3 and ROCO_2Li (main components of the outer-SEI) than for LiF (main component of the inner-SEI) [39-42]. This indicates that the lithium-ion transport through the inner-SEI is the major limiting factor in the case of Si electrodes.

Comment 13: *Most of the images contain too much information. Moreover, most of the 2D images (e.g. 5f, 5j, 6c-g, S3) are of bad quality; the range of the represented values and the image contrast is not well adapted, some of the images seem to be unsharp (e.g. 6 f,g,j). As such, it impossible to check the related conclusions of the authors. Just two examples; in lines 267-268, for Fig. 5b they provide “a roughness of about 2.4 nm”, in lines 257-259 in Fig. S3 it is “a roughness of about 1.2 nm”. Neither of these values can be “seen” from these figures. The authors should provide the appropriate mean values (roughness, thickness etc) and their statistical fluctuation instead of such approximate values.*

Answer: We thank the reviewer for bringing up these important points. We have adjusted the contrast of Fig. 5j-5n (previously Fig. 6c-6g) and Fig. S5 (previously Fig. S3). Additionally, they are replaced by new versions with improved graphic quality. We apologize for the blurriness in Figs. 5f, 5m (previously Fig. 6f), which were indeed caused by surface fluctuations as a result of the sudden growth of outer-SEI layer around the cracked area.

Regarding the estimation of the roughness for Fig. 5b and Fig. S5 (previously Fig. S3), the values reported in the paper, 2.4 nm and 1.2 nm, are calculated using the Nanoscope Analysis program [Author-reply-2-ref. 6]. The use of the word “about” was incorrect and has since been removed. It should be noted that the roughness provided in the paper is the Root Mean Square (RMS) average of the departure of the roughness profiles from the mean value. This has since been clarified in the manuscript. We agree that at first glance, with a height

variation from -5.5 nm to 6 nm (indicated by the scale bar in Author-reply-2-Fig. 5a), the roughness appears to be larger than the reported 2.4 nm. For that, we show the corresponding histogram in Author-reply-2-Fig. 5b. A Gaussian line fit yields a Full Width as Half Maximum of 5.33 nm, which in turn corresponds to an RMS roughness of $5.33/2\sqrt{2\ln(2)} = 2.26$ nm. The fitted value is in good agreement with the 2.4 nm given by the Nanoscope Analysis program, the slight discrepancy is attributed to the different methods of computation.

Author-reply-2-Fig. 5. (a) Same AFM topography result as Fig. 5b. (b) histogram of the height variation as shown in (a) and Gaussian fit.

[Author-reply-2-ref. 6] <http://nanoscaleworld.bruker-axs.com/nanoscaleworld/media/p/775.aspx>

Comment 14: Figure 6j is very confusing with two different X-axis, overlapping figure captions and X-axis title. Please include less information and re-organize this sub-figure.

Answer: We thank the reviewer for this excellent comment. Fig. 5q (previously Fig. 6j) has been restructured for clarity. Its X axis (previously the scanning position) has been converted into the scanning potential.

Comment 15: Lines 384-385: “The density of these early stage defects was low, making them essentially invisible for non-local techniques that average over the entire sample.” Could you please quantify what “low” means? Have you performed such non-local measurements? If yes, which ones and what was the analytical sensitivity?

Answer: To demonstrate that these early-stage defects would be essentially invisible for non-local techniques, we show results of rocking curves after the first 3 cycles (Author-reply-2-Fig.6). The rocking curve line profile is generated by integrating the intensity over a selected region of interest (RoI) on the 2D detector. Two RoIs were chosen, RoI1 ($25 \times 8 \mu\text{m}^2$) covers only the defective area while RoI2 ($430 \times 100 \mu\text{m}^2$) covers the entire field of view (FoV), as shown in Author-reply-2-Fig.6a.

Author-reply-2-Fig. 6. Rocking curve line profile generated by integrating intensity in RoI1 (b) and RoI2 (c) after cycle 1, 2 and 3. The definition of the RoIs is shown in (a).

The rocking curve line profile of RoI1 is different immediately after cycle 2. Higher intensity (marked by the red arrow in Author-reply-2-Fig.6b) was observed at about -0.03° off the Si (004) Bragg θ angle. The higher intensity was due to the scattering by the defects, and it was at this angle where all the *operando* dark field imaging (time scan) were carried out. Meanwhile, no changes were observed on the rocking curve line profile of RoI2 for the first 3 cycles, see Author-reply-2-Fig.6c. The integrated intensity over RoI2 represents what can be obtained with conventional XRD (one of the non-local techniques). While defects were already present inside RoI2 after the 2nd cycle, its integrated intensity showed little changes as it was dominated by scattering from the non-defective area. It is evident from this comparison that while conventional XRD is just as sensitive to lattice deformations as FFDXM, spatially resolved methods are required for detecting low density (early-stage) defects. The aforementioned figure and discussion have been added to the Supplementary Materials (Fig. S2).

Moreover, we have added information on the evolution of the surface area covered by the defects for cycle 1 to 6 (Author-reply-2-Fig. 7) to the Supplementary Materials as the new Fig. S3. The result echoes what was mentioned in the above paragraph. The early-stage defects covered a mere 0.3% of the entire surface area (<100 ppm of the volume considering an X-ray penetration depth of $200 \mu\text{m}$) and would hence be undetectable to non-local techniques such as conventional XRD. The defect coverage only became significant (rose to $\sim 20\%$) after 2000 seconds of accumulated holding at 5 mV.

Author-reply-2-Fig. 7. Potential, current and defect coverage plot for cycle 1-6.

Finally, we would like to reassure the reviewer that the structural defects described in the paper is not an isolated event. A surveyance scan covering $2 \times 1 \text{ mm}^2$ of the Si electrode shows that at the end of the 6th cycle, about 20% of the surface was consistently covered with these defects. The formation of the early-stage defects does not seem to be dependent on the nature of the charging/discharging either. We have performed FFDXM experiments on multiple samples with both cyclic voltammetry and galvanostatic cycling. Author-reply-2-Fig. 8 shows the result on another sample during galvanostatic charging and holding, extracted from FFDXM images. The onset and evolution of the defect density is comparable to what was observed with cyclic voltammetry. This last figure has also been added to the Supplementary Materials (new Fig. S8) and briefly mentioned in the discussion section of the manuscript: We note that the onset of the early-stage defects and their evolution have been consistently observed over repeated experiments, and under various cycling strategies (Fig. S8).

Author-reply-2-Fig. 8. Si electrode potential (E) and percentage of the surface area covered by defects as a function of time during 1st galvanostatic charging and constant voltage holding of another sample. The result is calculated from FFDXM measurements of an area of 430×100 μm.

Comment 16: Lines 443: “The rigidity of the thick Si substrates is necessary for high precision single-crystal diffraction measurements”. Does it mean that only thick (~200 μm) crystals can be studied with the method? This would strongly limit its possible application (e.g. excluding nanoparticles/thin-films?). Could you please comment on this? Could you please include a discussion on the limits of the experimental approach you used?

Answer: We thank the reviewer for this excellent comment. As stated previously in the answer to comment #5, high quality single crystals are ideal for the accurate measurements of defects with weak distortion and low density. While it is possible to study single crystal thin film electrodes of a few hundred nanometers thick, epitaxial thin films are desired over free standing ones. This is because free standing thin films (membranes) are often curved (bent) due to the absence of a rigid support, and easily strained by adhesives. These introduce inherent lattice tilt and strain in the sample which lead to longer measurement time (larger reciprocal space to cover), and in some cases difficulties in extracting quantitative results. It is worth mentioning that with the ongoing upgrade of various synchrotrons to Diffraction Limit Storage Rings (DLSR), it might be possible to one day image even thinner films (~ 50 nm). FFDXM, unfortunately, is not suitable for nanoparticle electrodes. This is because most nanoparticles are deposited with random crystallographic orientations (*i.e.* different nanoparticle would diffract at different theta, phi and psi angles). Being able to image simultaneously only one or two nanoparticles defeats the purpose of having a large FoV, in which case it becomes better to perform the experiment with a scanning diffraction X-ray microscope.

This discussion has briefly been mentioned in the methods section of the manuscript: *The rigidity of the thick Si substrates is necessary for high-precision measurements of the structural deformation. Free standing thin films are not recommended as they are often bent*

(inherent lattice tilt) due to the absence of a rigid support, and easily strained (inherent lattice strain) by adhesives.

Reviewers' Comments:

Reviewer #1:

Remarks to the Author:

I can see the authors have made quite some revision and it is now a much clearer and improved version. While I appreciate all their effort and the capability of FFDXM in detecting Si structural deformation as early as from the second cycle, I will have to mention the deformation and amorphization of Si at the first cycle has been well known by CV or other characterization methods. The authors have done good scientific research to visually observed the deformation using FFDXM while I have reservation on how the understanding will help to mitigate that. Because deformation and amorphization, which lead to defect formation, has long been known though it is not visually or as quantitative. I also would like to remind the authors that there are quite some mechanical study on Si deformation from Insun Yoon and Shuman Xia etc to detect the deformation of Si and the deformation can be observed at the first cycle.

Another question that I have is the role of SEI on Si deformation. I think this could be very good point and scientifically new to allow the paper to be published. However, I feel the current support is not convincing enough. I would like to suggest the authors to repeat the experiment in ether electrolyte or other electrolyte that is known to be stable against Si or metal to see if there will be any change to the "defect"/structural deformation. Although Si amorphization is also known to happen in ether electrolyte, inoic liquid or solid Li₂O (some in-situ TEM study), the "defect" formation may be different enough to study the role of SEI on Si structure change.

With that, I am not very enthusiastic for the paper to be published in current form but I am good if it were. I will be very excited if the authors do see different "defect" formation process between different electrolyte and elucidate the role of SEI. I believe that would be an exciting work to the community.

Reviewer #2:

Remarks to the Author:

The authors gave detailed reply to all of the questions and remarks of the referees. They have thoroughly rewritten the manuscript including new measurements and data strenthening the major claims of the paper. With these modifications I recommend the paper for publication.

Reviewers' comments:

Reviewer #1 (Remarks to the Author):

I can see the authors have made quite some revision and it is now a much clearer and improved version. While I appreciate all their effort and the capability of FFDXM in detecting Si structural deformation as early as from the second cycle, I will have to mention the deformation and amorphization of Si at the first cycle has been well known by CV or other characterization methods. The authors have done good scientific research to visually observed the deformation using FFDXM while I have reservation on how the understanding will help to mitigate that. Because deformation and amorphization, which lead to defect formation, has long been known thought it is not visually or as quantitative. I also would like to remind the authors that there are quite some mechanical study on Si deformation from Insun Yoon and Shuman Xia etc to detect the deformation of Si and the deformation can be observed at the first cycle.

Another question that I have is the role of SEI on Si deformation. I think this could be very good point and scientifically new to allow the paper to be published. However, I feel the current support is not convincing enough. I would like to suggest the authors to repeat the experiment in ether electrolyte or other electrolyte that is known to be stable against Si or metal to see if there will be any change to the "defect"/structural deformation. Although Si amorphization is also known to happen in ether electrolyte, inoic liquid or solid Li₂O (some in-situ TEM study), the "defect" formation may be different enough to study the role of SEI on Si structure change.

With that, I am not very enthusiastic for the paper to be published in current form but I am good if it were. I will be very excited if the authors do see different "defect" formation process between different electrolyte and elucidate the role of SEI. I believe that would be an exciting work to the community.

Reviewer #2 (Remarks to the Author):

The authors gave detailed reply to all of the questions and remarks of the referees. They have thoroughly rewritten the manuscript including new measurements and data strenthening the major claims of the paper. With these modifications I recommend the paper for publication.

Reviewers' comments:

Reviewer #1 (Remarks to the Author):

Comment 1: *I can see the authors have made quite some revision and it is now a much clearer and improved version. While I appreciate all their effort and the capability of FFDXM in detecting Si structural deformation as early as from the second cycle, I will have to mention the deformation and amorphization of Si at the first cycle has been well known by CV or other characterization methods. The authors have done good scientific research to visually observed the deformation using FFDXM while I have reservation on how the understanding will help to mitigate that. Because deformation and amorphization, which lead to defect formation, has long been known thought it is not visually or as quantitative. I also would like to remind the authors that there are quite some mechanical study on Si deformation from Insun Yoon and Shuman Xia etc to detect the deformation of Si and the deformation can be observed at the first cycle.*

Answer: We would like to point out that the early-stage defects (Fig. 2 of the manuscript) was observed under very limited amount of lithiation, achieved by the relatively high CV scan rate. In fact, for the first three cycles, the effective lithiation time (during which the potential stayed below 100 mV) is merely 38 sec per cycle, reaching roughly 0.1% - 0.2% of the full capacity. The limited amount of lithiation facilitates the observation of the defects when they were initially formed, which in turn allowed us to study their origin by quantitative measurements (Fig. 3). This was mentioned in our author response to previous Comment #1, and in the updated manuscript line 201 (**Results** section) and line 360 (**Discussion** section). In comparison, the AFM study, authored by Insun Yoon *et al.* [Author-reply-1-ref.1], has an effective lithiation time (by holding potential at 100 and 50 mV) of more than 3000 seconds in their first cycle. Similarly, the two studies co-authored [Author-reply-1-ref.2 and 3] by Shuman Xia *et al.* reached in their first cycle a lithiation capacity of 60 and 100%, respectively.

[Author-reply-1-ref. 1] I. Yoon, D. P. Abraham, B. L. Lucht, A. F. Bower, P. R. Guduru, In situ measurement of solid electrolyte interphase evolution on silicon anode using atomic force microscopy. *Adv. Energy Mater.* **2016**, *6*, 1600099.

[Author-reply-1-ref. 2] X. Wang, F. Fan, J. Wang, H. Wang, S. Tao, A. Yang, Y. Liu, H. B. Chew, S. X. Mao, T. Zhu, S. Xia, High damage tolerance of electrochemically lithiated silicon. *Nat. Commun.* **2015**, *6*, 8417.

[Author-reply-1-ref. 3] Y. He, F. Fan, S. Xia, Y. Yang, C. T. Harris, H. Li, J. Y. Huang, S. X. Mao, T. Zhu, Two-phase electrochemical lithiation in amorphous silicon, *Nano Lett.* **2013**, *13*, 709-716.

Comment 2: *Another question that I have is the role of SEI on Si deformation. I think this could be very good point and scientifically new to allow the paper to be published. However, I feel the current support is not convincing enough. I would like to suggest the authors to repeat the experiment in ether electrolyte or other electrolyte that is known to be stable against Si or metal to see if there will be any change to the "defect"/structural deformation.*

Although Si amorphization is also known to happen in ether electrolyte, ionic liquid or solid Li₂O (some in-situ TEM study), the "defect" formation may be different enough to study the role of SEI on Si structure change.

With that, I am not very enthusiastic for the paper to be published in current form but I am good if it were. I will be very excited if the authors do see different "defect" formation process between different electrolyte and elucidate the role of SEI. I believe that would be an exciting work to the community.

Answer: While we agree with the reviewer that the degree of SEI inhomogeneities and hence the defect formation would depend on the electrolyte choice [Author-reply-1-ref.4-6], we believe that the reviewer's question regarding *the role of SEI on Si deformation* was sufficiently answered by our example in Supplementary Figure 6. In that example, we coated the electrode with a homogeneous artificial SEI (20 nm Li₄Ti₅O₁₂ / 200 nm Li₃PO₄) and showed that no defect was observed under similar cycling conditions. Additionally, we hope that this last example answers another question the reviewer had (comment 1) as to *how the understanding (of our work) will help to mitigate that (early-stage defect formation in Si)*. Our work unveils the correlation between inhomogeneities in the SEI and defect formation in the Si electrode. We believe that by minimizing these inhomogeneities (*e.g.* using homogeneous artificial-SEI), one could mitigate defect formation in the Si electrode, thus improving the structural stability of Si-based LIBs.

We appreciate the time and effort the reviewer has invested in helping us improve the quality of our work.

- [Author-reply-1-ref. 4] Y. Zhang, N. Du, D. Yang, Design superior solid electrolyte interface on silicon anodes for high-performance lithium-ion battery, *Nanoscale*, **2019**, *11*, 19086-19104.
- [Author-reply-1-ref. 5] J. Zheng, H. Zheng, R. Wang, L. Ben, W. Liu, L. Chen, L. Chen, H. Li, 3D visualization of inhomogeneous multi-layered structure and Young's modulus of the solid electrolyte interphase (SEI) on silicon anodes for lithium ion batteries, *Phys. Chem. Chem. Phys.* **2014**, *16*, 13229-13238.
- [Author-reply-1-ref. 6] S. Benning, C. Chen, R. -A. Eichel, P. H. L. Notten, Direction observation of SEI formation and lithiation in thin-film silicon electrodes via in-situ electrochemical atomic microscopy, *ACS Appl. Energy Mater.* **2019**, *2*, 6791-6767.

Reviewer #2 (Remarks to the Author):

The authors gave detailed reply to all of the questions and remarks of the referees. They have thoroughly rewritten the manuscript including new measurements and data strengthening the major claims of the paper. With these modifications I recommend the paper for publication.

Answer: We are delighted to see the reviewer's recommendation for the publication of the manuscript. We would like to thank the reviewer once again for the time and efforts he/she has invested in helping us improve the quality of our work.